# A cascade model of dynamic cerebral autoregulation

Takuya Kurazumi[1,2,3] , Kartavya Sharma[2,4] , Ricardo R. J. Wennekers[1,5] , Tsubasa Tomoto[1,2,6] ,
Danilo Cardim[1,2] , Junyeon Won[1,2] , John Ashley[1,2] , Jurgen A. H. R. Claassen[5]
and Rong Zhang[1,2,7,8]

[1]*Institute for Exercise and Environmental Medicine, Texas Health Presbyterian Hospital Dallas, Dallas, TX, USA*

[2]*Department of Neurology, University of Texas Southwestern Medical Center, Dallas, TX, USA*

[3]*Department of Social Medicine, Division of Hygiene, Nihon University School of Medicine, Tokyo, Japan*

[4]*Department of Neurology, Mayo Clinic, Rochester, MN, USA*

[5]*Department of Geriatrics, Radboud University Medical Center, Nijmegen, the Netherlands*

[6]*Human Informatics and Interaction Research Institute, National Institute of Advanced Industrial Science and Technology, Ibaraki, Japan*

[7]*Department of Internal Medicine, University of Texas Southwestern Medical Center, Dallas, TX, USA*

[8]*Department of Biomedical Engineering, University of Texas Southwestern Medical Center, Dallas, TX, USA*

Handling Editors: Eleonora Grandi & Diego Candia-Rivera

The peer review history is available in the Supporting Information section of this article
(https://doi.org/10.1113/JP290519#support-information-section)

**Abstract figure legend** This study examined a two-component cascade model linking upstream dynamic cerebral auto-regulation (dCA) with downstream microvascular function (MF) to explain how blood pressure oscillations influence cortical oxygenation. Transfer function analysis in the frequency domain was applied to quantify $H_1(f)$ (MAP→CBFV; dCA), $H_2(f)$ (CBFV→O$_2$Hb; MF), and $H_0(f)$ (MAP→O$_2$Hb; empirical total response). The proposed cascade model ($H_c(f) = H_1(f) \times H_2(f)$) showed strong agreement with empirical transfer function in gain, phase and coherence across all frequency ranges at supine rest. Forced oscillations at 0.05 Hz induced by sit–stand manoeuvres further strengthened this

T. Kurazumi and K. Sharma contributed equally to this work.

The Journal of Physiology

agreement. These findings support the cascade model as a mechanistic framework linking arterial pressure fluctuations to cortical oxygenation and demonstrate its physiological relevance.

**Abstract** Cerebral blood flow is stabilized through dynamic adjustments across both macro- and microvascular compartments. While dynamic cerebral autoregulation (dCA) quantifies upstream pressure–flow coupling, downstream microvascular responses are less well characterized and may represent a distinct but functionally linked process. This study tested whether a two-component cascade model, treating dCA and an empirically derived index of microvascular function (MF) as sequential stages can represent the integrated regulation of cortical oxygenation. Data from 41 healthy adults (20–45 years) were analysed. Beat-to-beat mean arterial pressure (MAP), middle cerebral artery flow velocity (CBFV) and cortical oxyhaemoglobin ($O_2Hb$) were recorded during supine spontaneous oscillations and forced oscillations at 0.05 Hz using repeated sit–stand manoeuvres. Transfer function analysis quantified frequency-domain gain, phase and coherence for MAP→CBFV (dCA), CBFV→$O_2Hb$ (MF) and MAP→$O_2Hb$ (total pathway). The cascade model was computed as the product of dCA and MF transfer functions. The cascade model derived indices showed strong correlations with total pathway gain, phase and coherence measures during both spontaneous and forced oscillations, with improved linear coupling under forced oscillations. These results support the applicability of a two-component cascade model for integrated cerebrovascular regulation and suggest that serial interactions between macrovascular and microvascular regulatory mechanisms jointly shape the frequency-dependent propagation of arterial pressure to brain-tissue oxygenation dynamics.

(Received 14 November 2025; accepted after revision 7 April 2026; first published online 23 April 2026)

**Corresponding author** R. Zhang: Institute for Exercise and Environmental Medicine, Texas Health Presbyterian Hospital Dallas, Dallas, TX 75231, USA. Email: RongZhang@TexasHealth.org

## Key points

- Cerebral blood flow is stabilized by coordinated regulation across large arteries and microvessels, but their dynamic interaction has not been experimentally modelled.
- We examined whether a two-component cascade model linking upstream dynamic cerebral auto-regulation and downstream microvascular function can explain how blood pressure oscillations influence cortical oxygenation.
- In 41 healthy adults, we recorded beat-to-beat blood pressure, middle cerebral artery blood flow velocity and near-infrared spectroscopy-derived oxygenation during rest and sit–stand manoeuvres, and analysed them using transfer function analysis.
- Gains, phases and coherences derived from the cascade model closely matched those from the direct blood pressure–$O_2Hb$ relationship, particularly under forced oscillations during sit–stand, demonstrating the model's physiological relevance.
- The cascade model provides a mechanistic framework to separate and quantify large- and small-vessel contributions to cerebral blood flow regulation, with potential application in future studies of ageing and cerebrovascular disease.

## Introduction

Despite constant fluctuations in systemic blood pressure, cerebral blood flow (CBF) is stabilized through coordinated responses across the vascular tree, from the proximal extracranial large arteries to the arterioles and the microcirculation (De Silva & Faraci, 2016; Gould et al., 2017; Iadecola, 2017). This regulatory capacity, termed cerebral autoregulation (CA), is described both by its steady-state effectiveness across a range of pressures (static CA) and by its ability to buffer transient blood pressure fluctuations (dynamic CA) (Claassen et al., 2021; Lassen, 1964; Panerai, 1998).

Dynamic CA (dCA) is commonly studied using transfer function analysis (TFA) of continuous changes in mean arterial pressure (via finger-clamp photo-

plethysmography) and cerebral blood flow velocity (CBFV) in the middle cerebral artery (MCA) measured with transcranial Doppler ultrasound (TCD), assuming that velocity changes proportionally reflect CBF changes when arterial diameter is constant (Claassen et al., 2016; Panerai et al., 2023; Zhang et al., 1998). Although TCD primarily measures blood flow velocity in the large conduit arteries, these signals are likely affected by downstream vascular resistance and compliance. Consequently, conventional TCD-based TFA reflects the composite influence of both macrovascular and microvascular dynamics without distinguishing their individual contributions. However, previous studies have shown that macro- and microvascular haemodynamic regulation may diverge under specific physiological and pathological conditions (Favilla et al., 2023; Lim et al., 2024; Müller et al., 2020; Shoemaker et al., 2023). This gap motivates further inquiry to explicitly delineate the interplay between the upstream macrovascular and downstream microvascular regulation.

Near-infrared spectroscopy (NIRS) is a non-invasive optical method for assessing cerebral tissue oxygenation and haemodynamics, primarily reflecting micro-circulatory perfusion (Kainerstorfer et al., 2015). Unlike TCD, which indexes blood flow velocity in large arteries, NIRS measures dynamic fluctuations in oxyhaemoglobin ($O_2Hb$) and deoxyhaemoglobin (HHb) within cortical microvasculature, encompassing arterioles, venules and capillaries (Owen-Reece et al., 1999). These signals reflect changes in both local tissue oxygenation and micro-vascular blood volume, offering a marker of downstream microvascular responses to changes in upstream blood flow.

Prior work has shown coupling between $O_2Hb$ fluctuations and upstream CBFV, suggesting a link between macro- and microvascular blood flow regulation (Müller et al., 2020; Tarumi et al., 2014; van Beek et al., 2012). These downstream $O_2Hb$ responses to changes in upstream blood flow – which we henceforth refer to as 'microvascular function' (MF) – may provide complementary insights into dCA (Elting et al., 2020).

Theoretically, dCA derived from TCD and MF can be conceptualized as sequential regulatory components of the cerebral circulation. A cascade modelling approach formalizes this integration by treating the blood pressure (BP)→CBFV relationship and the CBFV→$O_2Hb$ relationship as serial transfer functions, whose product represents the overall BP→$O_2Hb$ relationship (Fig. 1). Conceptual parallels exist in cardiovascular physiology. For example, Ikeda et al. (1996) demonstrated that the arterial baroreflex loop can be decomposed into neural and peripheral arcs and modelled as a series system, allowing physiologically meaningful interpretation using TFA (Ikeda et al., 1996); Shibata et al. (2006) demonstrated that ventriculo–arterial coupling followed by the baroreflex control could be represented as a cascade model whose combined gain predicted blood pressure–heart rate variability (Shibata et al., 2006); while Hieda et al. (2019) extended this approach to show that exercise training enhances cardiovascular regulation by strengthening the sequential control steps of a three-component cascade (Hieda et al., 2019). Extending this framework to cerebral autoregulation, we propose that a cascade model integrating dCA and MF may capture the underlying regulatory mechanisms of BP–$O_2Hb$ dynamics, providing a physiologically grounded framework for quantifying integrated CBF regulation across the cerebral vascular compartments.

## Methods

**Participants.** Participants were originally recruited as part of an Institutional Review Board-approved study investigating normal ageing and brain vascular function using flyers and newspaper advertisements in the Dallas-Fort Worth metroplex (Tarumi et al., 2022; Tomoto et al., 2020; Xing et al., 2017). Data from

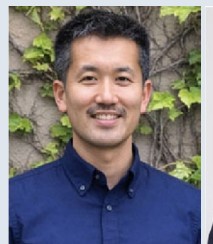
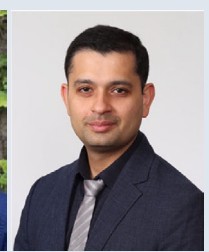

**Takuya Kurazumi** is a physician specializing in anaesthesiology and intensive care medicine, with additional expertise in space medicine. He received his MD from Gunma University School of Medicine in 2008 and his PhD from Keio University Graduate School of Medicine in 2014. His research focuses on cerebrovascular and intracranial physiology, with particular interest in cerebral autoregulation and systemic–cerebral interactions. He is also engaged in environmental medicine and ageing-related research, exploring how physiological regulation adapts to environmental and age-related challenges. **Kartavya Sharma** is an Associate Professor of Neurology at Mayo Clinic, Rochester. He completed medical training at the All India Institute of Medical Sciences, New Delhi, followed by neurology residency at UT Southwestern and a neurocritical care fellowship at Johns Hopkins University. His research focuses on neurovascular interactions underlying recovery from severe brain injury, using multimodal approaches integrating electrophysiology, optical imaging and cerebrovascular physiology. His work aims to develop biomarkers of brain function to better understand recovery and improve care for patients with disorders of consciousness.

*A*

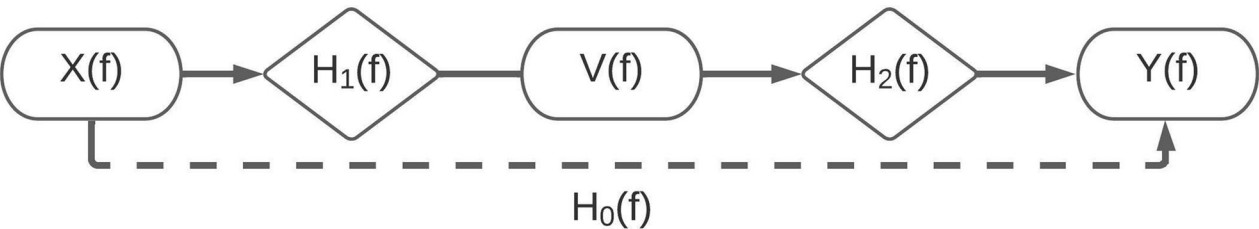

*B*

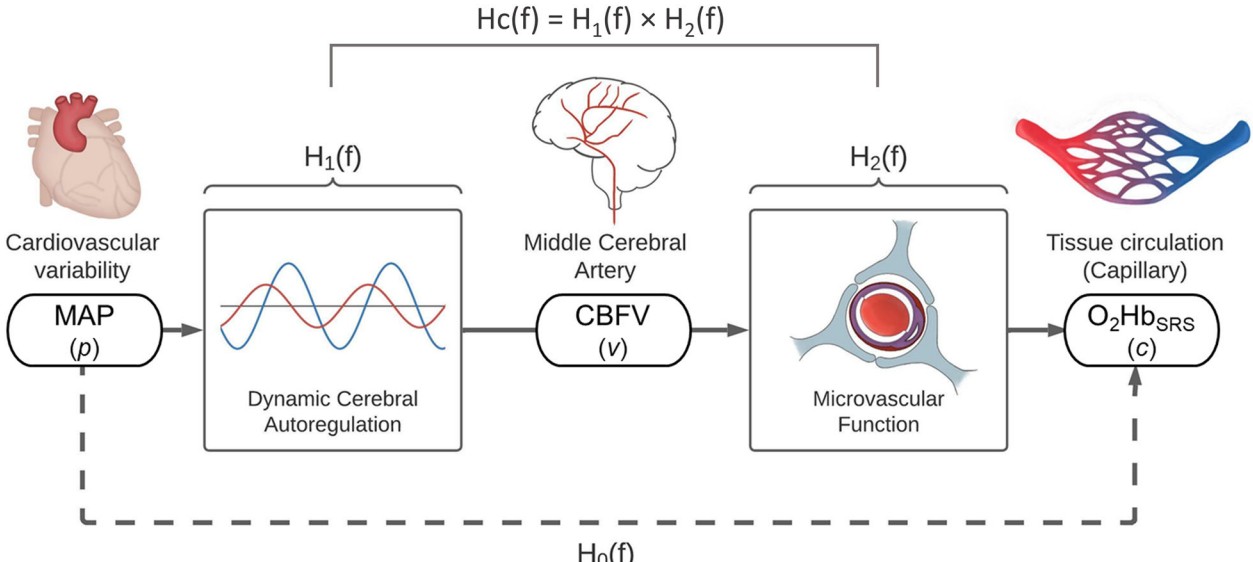

**Figure 1. Schematic representation of the cascade model for assessing multiscale cerebrovascular regulation**

*A*, generic transfer-function framework illustrating sequential transformations between an input signal $X(f)$, an intermediate variable $V(f)$ and an output signal $Y(f)$ through frequency-domain responses $H_1(f)$ and $H_2(f)$. The overall system response $H_0(f)$ represents the composite transfer from $X(f)$ to $Y(f)$. *B*, cascade model for integrated dynamic regulation of the cerebral macro- and microcirculation. Mean arterial pressure (MAP, $p$) serves as the input signal ($X(f)$), cerebral blood flow velocity (CBFV, $v$) as the intermediate variable ($V(f)$), and oxyhaemoglobin ($O_2Hb_{SRS}$, $c$) as the final output ($Y(f)$). The modelled cascade ($H_c(f) = H_1(f) \times H_2(f)$) represents the serial interaction between dynamic cerebral autoregulation ($H_1(f)$), modulating the pressure–flow relationship, and microvascular function ($H_2(f)$), governing local oxygenation dynamics. The empirical total function ($H_0(f)$) reflects the directly measured MAP–$O_2Hb_{SRS}$ relationship. The heart, middle cerebral artery and capillary bed images schematically represent the source of systemic arterial pressure fluctuations (MAP), the large-vessel pressure–flow compartment ($H_1(f)$) and the downstream cerebral microcirculation governing flow-oxygenation dynamics ($H_2(f)$), respectively. The neurovascular unit schematic in $H_2(f)$ is conceptual and does not depict specific cellular mechanisms in the model.

41 healthy young adults (aged 20–45 years) from this previously established cohort were analysed retrospectively in the present study. Exclusion criteria were: (1) heart disease (screened by 12-lead ECG and echocardiography); (2) BP >140/90 mmHg measured by ambulatory monitoring; (3) carotid artery atherosclerotic plaque or stenosis with >50% occlusion detected by ultrasound; (4) body mass index >40 kg/m$^2$; (5) current or recent smoking (within past 2 years); and (6) alcohol or drug abuse.

All study procedures were approved by the Institutional Review Boards of the University of Texas Southwestern Medical Center and Texas Health Presbyterian Hospital Dallas (STU 102010-069). The study was conducted in accordance with the principles of the *Declaration of Helsinki* (except for registration in a database) and the Belmont Report. Written informed consent was obtained from all participants. Deidentified research data can be shared with qualified investigators upon request with a data transfer agreement.

**Instrumentation and measurement.** Continuous cerebral tissue oxygenation was measured using a NIRS device (NIRO-200NX, Hamamatsu Photonics, Hamamatsu, Japan) with three wavelengths (735, 810 and 850 nm), configured to output NIRS-derived variables at 5 Hz. Optodes were mounted on the left forehead using the manufacturer's standard adult probe with a fixed 4 cm emitter–detector separation. The system employs an internal spatially resolved spectroscopy (SRS) algorithm that uses multiple photodetectors within the probe housing to estimate the tissue oxygenation index (TOI) and the normalized total haemoglobin index (nTHI). TOI reflects the fraction of oxygenated to total haemoglobin, while nTHI represents relative changes in total haemoglobin concentration normalized to baseline.

The device also outputs relative concentration changes in oxyhaemoglobin ($\Delta O_2Hb$), deoxyhaemoglobin ($\Delta HHb$) and total haemoglobin ($\Delta tHb$) based on the modified Beer–Lambert law (MBLL) method (Delpy et al., 1988), using the manufacturer's recommended differential pathlength factor of 5.93 for the adult forehead. However, in the present study, we focused on the SRS-derived indices (TOI, nTHI) rather than the MBLL-derived variables, guided by prior work indicating that SRS-based measures are less influenced by extracerebral contamination and may better represent cortical microvascular oxygenation (Al-Rawi et al., 2001; Suzuki et al., 1999; Terborg et al., 2003). The computation of SRS-derived oxyhaemoglobin ($O_2Hb_{SRS}$) from these indices is described below.

Continuous CBFV in the MCA was measured using TCD ultrasonography (Multi-Dop X2, Compumedics/DWL, Singen, Germany), where CBFV represents the velocity of blood flow within the blood vessel rather than volumetric cerebral blood flow (Nichols et al., 2022; White, 2011). A 2-MHz probe was placed over the right temporal region and was securely attached by an individually created mould to fit the facial bone structure and keep the position and angle of the probe unchanged during the study (Giller & Giller, 1997). The sampling depth ranged from 42 to 55 mm and the angle was adjusted to optimize the signal quality for each participant using standard procedure (Aaslid et al., 1982). The Doppler signal was acquired at 100 Hz, enabling high temporal resolution and beat-to-beat assessment of CBFV.

Continuous BP was measured from the left middle finger at the level of the participant's heart using finger-clamp photoplethysmography (Finapres 2300, Ohmeda Monitoring Systems, Englewood, CO, USA). The brachial BP at the level of the heart was measured intermittently by an electro sphygmomanometer (Suntech, Morrisville, NC, USA) on the right upper arm at the beginning of each baseline period. End-tidal carbon dioxide ($EtCO_2$) was measured using capnography (Carpnogard, Novametrix, Wallingford, CT, USA). Arterial blood oxygen saturation ($SpO_2$) was measured by a pulse oximeter (Biox 3700, Ohmeda Monitoring Systems, Boulder, CO, USA). Heart rate (HR) was determined by a three-lead ECG system (Hewlett-Packard, Palo Alto, CA, USA). NIRS signals (output at 5 Hz by the NIRO system) and all other physiological signals were digitized by the BIOPAC data acquisition system at 250 Hz and stored for offline analysis using Acknowledge software (BIOPAC Systems, Goleta, CA, USA).

**Study protocol.** The experimental protocol consisted of two conditions: a resting baseline period (supine) and repeated sit–stand manoeuvres, performed on separate visit days. Participants were instructed to refrain from high-intensity exercise, caffeine and alcohol for at least 24 h prior to testing (Panerai et al., 2023; Tarumi et al., 2022; Xing et al., 2017). All measurements were conducted in an environmentally controlled laboratory with an ambient temperature of ~22°C. After a 20 min period for instrument setup and haemodynamic stabilization, participants rested supine for 10 min. Data were then extracted from the last 5 min of a physiologically stable segment within the 10 min supine rest period, based on signal stability and absence of artifacts. The repeated sit–stand manoeuvres consisted of 10 s sitting and 10 s standing cycles performed continuously for 5 min, corresponding to a frequency of 0.05 Hz. Participants rested quietly in a seated upright position for 20 min before beginning the manoeuvres, and an investigator provided verbal guidance throughout. The manoeuvres are a non-pharmacological method to induce

large changes in BP and CBFV for improving the signal-to-noise ratio and reliability of transfer function analysis (Claassen et al., 2009).

### Data analysis

*Preprocessing.* Beat-to-beat values of MAP, HR, CBFV, TOI and nTHI were analysed. R-wave peaks were detected from the ECG, and each cardiac cycle was defined by consecutive R–R intervals. Beat-to-beat values of MAP, HR, CBFV, TOI and nTHI were obtained by integrating each signal within the corresponding R–R interval to yield a single value per cardiac cycle. Cycle averaging was performed using AcqKnowledge software (BIOPAC Systems), according to previously described methods (Zhang et al., 1998). Breath-by-breath $EtCO_2$ was recorded simultaneously with MAP, HR, CBFV, TOI and nTHI. Fluctuations in oxyhaemoglobin are widely used as indicators of changes in regional blood flow and blood volume (Claassen et al., 2006; Owen-Reece et al., 1999). Accordingly, $O_2Hb_{SRS}$ was computed from the device-reported TOI and nTHI values based on the volume fraction principle (Lehmann et al., 1996; Suzuki et al., 1999):

$$O_2Hb_{SRS(t)} = TOI_{(t)} \times nTHI_{(t)} \qquad (1)$$

where $t$ indicates a given time point. This expression defines an index of relative cortical $O_2Hb$ changes, reflecting fluctuations in microvascular oxygenation and blood volume. Because both TOI and nTHI are dimensionless quantities derived from spatially resolved light attenuation, $O_2Hb_{SRS}$ is expressed in arbitrary units.

*Transfer function analysis.* Beat-to-beat means of MAP, CBFV and $O_2Hb_{SRS}$ were computed. Changes in CBFV and $O_2Hb_{SRS}$ were expressed as percentages of mean for spectral and transfer function analyses. All beat-to-beat signals and breath-by-breath $EtCO_2$ were linearly interpolated and resampled at 2 Hz to obtain equidistant time series. This sampling rate is sufficient to capture cerebrovascular dynamics up to 0.5 Hz (Nyquist frequency 1 Hz), and is consistent with our prior studies in healthy volunteers using similar experimental paradigms and datasets (Tarumi et al., 2014, 2022; Xing et al., 2017). The resampled signals were detrended with a third-order polynomial. Power spectral density (PSD) and transfer function estimates were computed using the Welch method. Data were segmented into 256-point windows with 50% overlap, and a Hanning window was applied. PSD was calculated by fast Fourier transform, and transfer functions were estimated by cross-spectral analysis between input and output signals with a spectral resolution of 0.0078 Hz (Iwasaki et al., 2011; Shibata et al., 2006; Tarumi et al., 2014). Analyses were performed in DADiSP/2002 (DSP Development Corporation, Newton, MA, USA).

The transfer function $H(f)$ between the input and output signals was defined as:

$$H(f) = \frac{S_{xy}(f)}{S_{xx}(f)} \qquad (2)$$

where $S_{xy}(f)$ is the cross-spectrum of input and output and $S_{xx}(f)$ is the auto-spectrum of the input.

From this, transfer function gain $|H(f)|$, phase $\Phi(f)$ and mean-squared coherence $MSC(f)$ were derived (Claassen et al., 2016; Panerai et al., 2023; Zhang et al., 1998). The gain quantifies the relative amplitude of output to input oscillations (i.e. the magnitude of changes in CBFV or $O_2Hb_{SRS}$ in response to changes in blood pressure or blood flow). Higher gain values indicate reduced attenuation of oscillations transmitted from the input to the output, whereas lower gain values indicate greater attenuation. The phase describes the temporal offset between the input and output, reflecting the time constant and/or delay of the vascular adjustment in response to changes in blood pressure or blood flow. A positive phase may indicate that changes in the output lead changes in the input, whereas a negative phase indicates that the output lags the input; phase values near zero indicate minimal temporal offset and greater synchrony between the input and output signals. The coherence represents the consistency of a linear relationship between the input and output across frequencies and is used to assess the reliability of gain and phase estimates; accordingly, gain and phase were interpreted only within the frequency ranges exhibiting sufficiently high coherence.

*Cascade model.* The cascade model was formulated as a series combination of transfer functions (Fig. 1). Transfer function from MAP ($p$) to CBFV ($v$) was defined as $H_1(f)$, from CBFV to $O_2Hb_{SRS}$ ($c$) as $H_2(f)$, and from MAP to $O_2Hb_{SRS}$ as $H_0(f)$:

$$H_1(f) = \frac{S_{pv}(f)}{S_{pp}(f)} \qquad (3a)$$

$$H_2(f) = \frac{S_{vc}(f)}{S_{vv}(f)} \qquad (3b)$$

$$H_0(f) = \frac{S_{pc}(f)}{S_{pp}(f)} \qquad (3c)$$

where $S_{pv}(f)$, $S_{vc}(f)$ and $S_{pc}(f)$ denote the cross-spectra between MAP and CBFV, CBFV and $O_2Hb_{SRS}$, and MAP and $O_2Hb_{SRS}$, respectively, and $S_{pp}(f)$ and $S_{vv}(f)$ denote the corresponding auto-spectra of MAP and CBFV.

For each transfer function, the gain, phase and coherence indices were derived: $G_1$, $P_1$, $C_1$ for

MAP→CBFV; $G_2$, $P_2$, $C_2$ for CBFV→O$_2$Hb$_{SRS}$; and $G_0$, $P_0$, $C_0$ for MAP→O$_2$Hb$_{SRS}$ (Claassen et al., 2016; Panerai et al., 2023; Zhang et al., 1998). Thus, $H_1(f)$ describes the upstream macrovascular CBFV responses to changes in blood pressures, $H_2(f)$ describes the downstream O$_2$Hb$_{SRS}$ responses to changes in upstream CBFV (i.e. MF), and $H_0(f)$ describes the integrated vascular responses.

The two-component cascade transfer function was obtained by combining $H_1(f)$ and $H_2(f)$, defined as (Smith, 2007):

$$H_c(f) = H_1(f) \times H_2(f) \qquad (4)$$

The cascade model gain equals the product of the individual component gains (Shibata et al., 2006; Smith, 2007):

$$G_c(f) = |H_1(f)| \times |H_2(f)| \qquad (5)$$

The cascade model phase is the sum of the individual component phases, reflecting accumulation of time delays. To avoid 'phase wrapping', negative phase values below 0.10 Hz during rest were excluded from the phase calculation (Claassen et al., 2016; Claassen et al., 2021; Panerai et al., 2023):

$$P_c(f) = \Phi 1(f) + \Phi 2(f) \qquad (6)$$

The cascade model coherence was estimated as the product of the component coherences (Menčík, 2016):

$$C_c(f) = C_1(f) \times C_2(f) \qquad (7)$$

Comparison of the cascade model derived $G_c(f)$, $P_c(f)$ and $C_c(f)$ with those derived from $H_0(f)$ provides a framework to test whether the cascade model represents the integrated BP–O$_2$Hb relationship (Hieda et al., 2019; Shibata et al., 2006).

*Cerebrovascular variability.* The PSDs of MAP, CBFV and O$_2$Hb$_{SRS}$, as well as their transfer function indices, were computed across very low (VLF: 0.02–0.07 Hz), low (LF: 0.07–0.20 Hz) and high (HF: 0.20–0.35 Hz) frequency bands during spontaneous oscillations at rest. These frequency bands are thought to reflect distinct vascular regulatory mechanisms underlying dynamic CA and MF (Claassen et al., 2009; Tomoto et al., 2021; Xing et al., 2017). During repeated sit–stand manoeuvres, the periodic response at 0.05 Hz was represented by averaging power spectral and transfer function values at the two adjacent frequencies (0.046875 and 0.0546875 Hz), as in prior studies (Tarumi et al., 2022; Tomoto et al., 2021). The PSD of HR and breath-by-breath EtCO$_2$ were also calculated to reveal its spectral power distribution in the frequency range of 0.0–0.5 Hz.

*Critical coherence threshold and coupling assessment.* Signal coupling strength was assessed using magnitude-squared coherence for each subject. Significant coupling was defined when the coherence exceeded the critical threshold of 0.34, corresponding to the 95% confidence limit for five overlapping spectral windows, as recommended by the Cerebrovascular Research Network (CARNet) white paper (Panerai et al., 2023). The number and percentage of subjects exhibiting significant coupling were then calculated for each pathway ($H_1$: MAP→CBFV; $H_2$: CBFV→O$_2$Hb$_{SRS}$; $H_0$: MAP→O$_2$Hb$_{SRS}$) and each frequency range (VLF, LF, HF during rest and at 0.05 Hz during sit–stand).

**Statistical analysis.** Data are presented as means ± standard deviation (SD). Group differences between women and men were assessed using an unpaired Student's $t$ test. Pearson correlation coefficients were calculated between $H_c(f)$ and $H_0(f)$ for transfer function gain, phase and coherence within the VLF, LF and HF ranges during rest, as well as at 0.05 Hz during sit–stand manoeuvres. The coefficients of determination ($R^2$) were obtained from linear regression analysis. A $P$-value of $<0.05$ was considered statistically significant. All statistical analyses were conducted in SPSS Statistics 20.0 (IBM Corp., Armonk, NY, USA).

## Results

**Participants' characteristics and cerebral haemodynamics during rest and sit–stand manoeuvres.** The mean haemodynamic data during the rest period and sit–stand manoeuvres as well as the participants' baseline characteristics are shown in Table 1. CBFV was significantly higher in women than men during both rest and sit–stand conditions. In contrast, TOI was significantly higher in men during the sit–stand manoeuvres. No sex differences were found in nTHI at either condition. EtCO$_2$ was significantly higher in men than in women during the sit–stand manoeuvres. HR increased during sit–stand manoeuvres ($P < 0.001$, paired $t$ test) while EtCO$_2$ did not differ between rest and sit–stand conditions ($P = 0.074$).

**Dynamic CA and MF during rest.** Group means and standard deviations (SD) for the PSDs of MAP, CBFV and O$_2$Hb$_{SRS}$, and dCA and MF TFA indices during rest are shown in Table 2. In men, spectral power of CBFV in the VLF band, and O$_2$Hb$_{SRS}$ in the HF band were higher than those in women. Group-averaged PSDs for MAP, CBFV and O$_2$Hb$_{SRS}$ are shown in Fig. 2A–C. Group-averaged PSDs for HR and EtCO$_2$ during rest are shown in the Appendix, Fig. A1.

The group-averaged transfer function gain spectra from MAP to CBFV ($G_1$) and from CBFV to O$_2$Hb$_{SRS}$ ($G_2$) are shown in Fig. 2D and E. No sex differences were observed

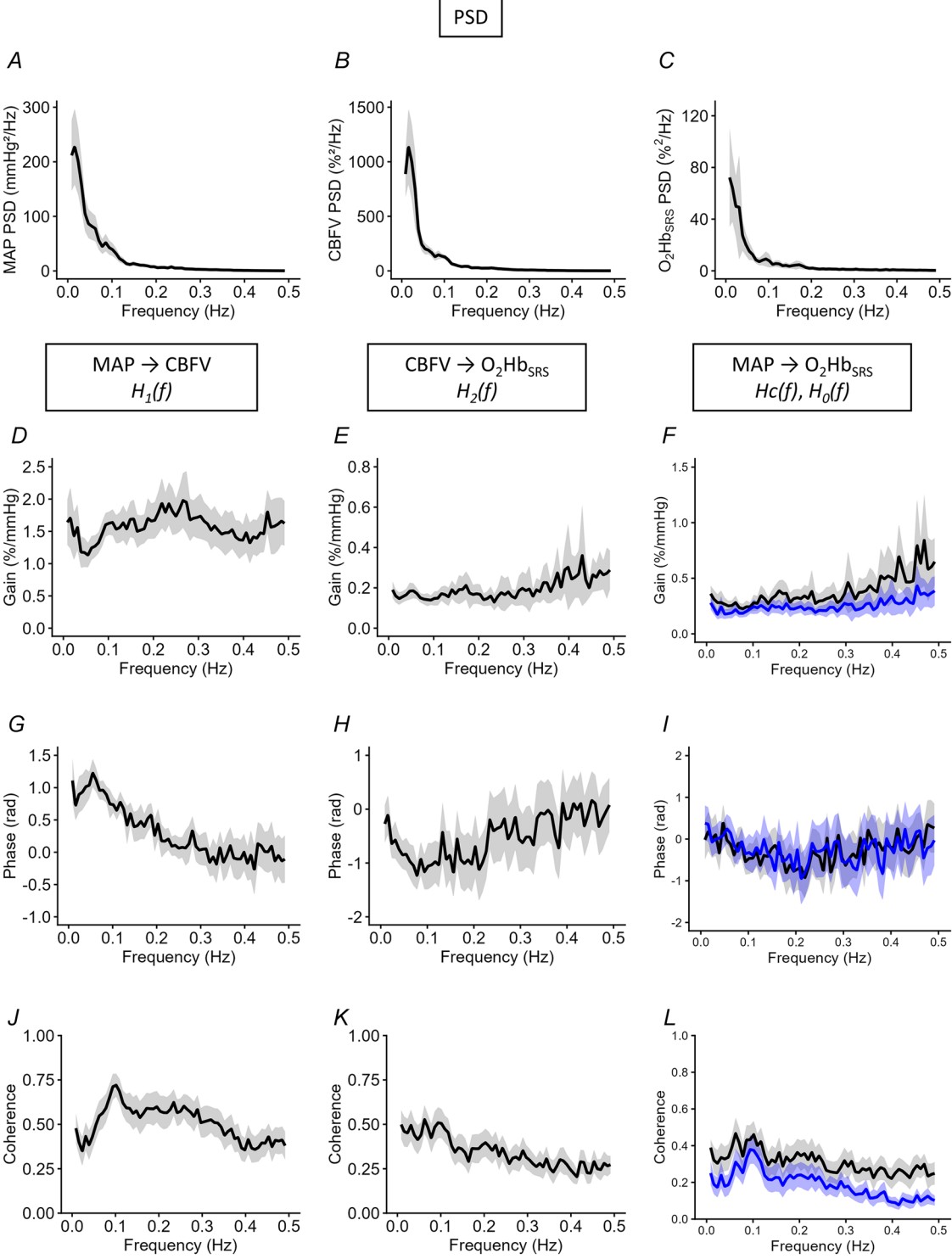

**Figure 2. Group-averaged power density spectral density and transfer function metrics during rest**
*A–C*, PSD of MAP, CBFV and $O_2Hb_{SRS}$, respectively. *D–F*, gain spectra of transfer functions between MAP→CBFV $(G_1)$, CBFV→$O_2Hb_{SRS}$ $(G_2)$, the modelled cascade $(G_c = G_1 \times G_2)$ and empirical total function MAP→$O_2Hb_{SRS}$ $(G_0)$. *G–I*, corresponding phase spectra for MAP→CBFV $(P_1)$, CBFV→$O_2Hb_{SRS}$ $(P_2)$, cascade model $(P_c)$ and total MAP→$O_2Hb_{SRS}$ $(P_0)$; *J–L*, Coherence functions for MAP→CBFV $(C_1)$; CBFV→$O_2Hb_{SRS}$ $(C_2)$ and the cascade $(C_c)$ and total MAP→$O_2Hb_{SRS}$ $(C_0)$. Thick lines denote the group means; grey shaded areas represent 95% confidence intervals across subjects. The cascade model $(H_1(f) \times H_2(f))$ is shown in blue. CBFV, cerebral blood flow velocity; MAP, mean arterial pressure; $O_2Hb_{SRS}$, oxyhaemoglobin calculated by the Spatially Resolved Spectroscopy method; PSD, power spectral density.

**Table 1. Participant characteristics and cerebral haemodynamics during rest and sit–stand manoeuvres**

| | Total | Women | Men | P |
|---|---|---|---|---|
| **Baseline characteristics** | | | | |
| n | 41 | 25 | 16 | |
| Age (years) | 33 (7) | 33 (7) | 34 (7) | 0.477 |
| Height (cm) | 169 (9) | 163 (6) | 177 (7) | <0.001 |
| Weight (kg) | 69 (14) | 62 (11) | 80 (10) | <0.001 |
| BMI (kg/m$^2$) | 24 (4) | 23 (3) | 26 (4) | 0.043 |
| SpO$_2$ (%) | 99 (1) | 99 (1) | 98 (1) | <0.001 |
| **Rest** | | | | |
| n | 41 | 25 | 16 | |
| MAP (mmHg) | 91 (13) | 93 (13) | 88 (14) | 0.230 |
| CBFV (cm/sec) | 60 (10) | 64 (11) | 55 (7) | 0.006 |
| TOI (a.u.) | 68 (6) | 67 (6) | 70 (7) | 0.151 |
| nTHI (a.u.) | 1.00 (0.03) | 1.00 (0.02) | 0.99 (0.04) | 0.610 |
| EtCO$_2$ (mmHg) | 38.6 (4.5) | 37.7 (4.8) | 39.9 (3.7) | 0.133 |
| HR (bpm) | 69 (12) | 71 (12) | 67 (12) | 0.274 |
| **Sit–stand** | | | | |
| n | 36 | 23 | 13 | |
| MAP (mmHg) | 100 (11) | 101 (11) | 99 (13) | 0.520 |
| CBFV (cm/s) | 54 (11) | 56 (11) | 48 (8) | 0.049 |
| TOI (a.u.) | 66 (6) | 65 (5) | 69 (6) | 0.027 |
| nTHI (a.u.) | 1.06 (0.06) | 1.05 (0.06) | 1.07 (0.05) | 0.385 |
| EtCO$_2$ (mmHg) | 37.1 (3.6) | 36.0 (3.6) | 39.1 (2.5) | 0.009 |
| HR (bpm) | 83 (8) | 82 (9) | 85 (7) | 0.258 |

Values are means (SD). *P*-values refer to comparisons between men and women. BMI, body mass index; CBFV, cerebral blood flow velocity; EtCO$_2$, partial pressure of end-tidal carbon dioxide; HR, heart rate; MAP, mean arterial pressure; nTHI, normalized total haemoglobin index; SpO$_2$, oxygen saturation; TOI, tissue oxygenation.

for $G_1$ (Table 2). However, $G_2$ was higher in the VLF band and was lower in the HF band in women than men. Group means of total gain ($G_0$) and cascade model gain ($G_c$) are shown in Fig. 2*F*, where $G_c$ followed a similar pattern to $G_0$, but it appears to be lower than $G_0$ at frequencies above 0.2 Hz.

Phase spectra are shown in Fig. 2*G–I*. MAP−CBFV phase ($P_1$) was positive and decreased with frequency, while CBFV−O$_2$Hb$_{SRS}$ phase ($P_2$) was negative across VLF, LF and HF bands. Negative phase values in the MAP–CBFV relationship occurred primarily at the lowest frequencies (∼40% of subjects at 0.0078 Hz) and were rare near 0.10 Hz (<5%), summarized in the Appendix, Table A1. No sex differences were observed in either $P_1$ or $P_2$. The total phase ($P_0$) and cascade model phase ($P_c$) showed similar patterns across frequency ranges (Fig. 2*I*).

Coherence spectra are shown in Fig. 2*J–L*. MAP–CBFV coherence ($C_1$) was ∼0.6 in the LF band, and >90% of subjects exceeded the critical coherence threshold of 0.34 (Table 3). CBFV–O$_2$Hb$_{SRS}$ coherence ($C_2$) was ∼0.5 in the VLF band and decreased at higher frequencies, although >60% of subjects surpassed the critical coherence in the VLF and LF ranges. The total coherence ($C_0$) and cascade model coherence ($C_c$) showed similar patterns across frequency ranges (Fig. 2*L*).

**Dynamic CA and MF during sit–stand manoeuvres.** Group means and standard errors for spectral power of MAP, CBFV and O$_2$Hb$_{SRS}$, as well as dCA and MF TFA indices during sit–stand manoeuvres are shown in Table 4. No sex differences were observed in spectral power of MAP, CBFV and O$_2$Hb$_{SRS}$. As expected, spectral power in MAP, CBFV and O$_2$Hb$_{SRS}$ increased significantly at 0.05 Hz during the manoeuvres (Fig. 3*A–C*). The full frequency spectra for these signals are provided in the Appendix, Fig. A2, for reference. Group-averaged PSDs for HR and EtCO$_2$ during sit–stand manoeuvres are shown in the Appendix, Fig. A1.

Group-averaged spectra for transfer function gain are shown in Fig. 3*D* and *E*. Gain from MAP to CBFV ($G_1$) was higher in men than women (Table 4), while gain from CBFV to O$_2$Hb$_{SRS}$ ($G_2$) was higher in women. The total gain from MAP to O$_2$Hb$_{SRS}$ ($G_0$) and cascade model gain ($G_c$) overlapped at 0.05 Hz (Fig. 3*F*).

Phase spectra are shown in Fig. 3*G–I*. MAP–CBFV phase ($P_1$) was positive at 0.05 Hz, while CBFV–O$_2$Hb$_{SRS}$ phase ($P_2$) was negative. The total phase ($P_0$) and cascade phase ($P_c$) overlapped at 0.05 Hz (Fig. 3*I*).

Coherence spectra are shown in Fig. 3*J–L*. MAP–CBFV coherence ($C_1$) and CBFV–O$_2$Hb$_{SRS}$ coherence ($C_2$) both exceeded 0.9 at 0.05 Hz, and all subjects exceeded the

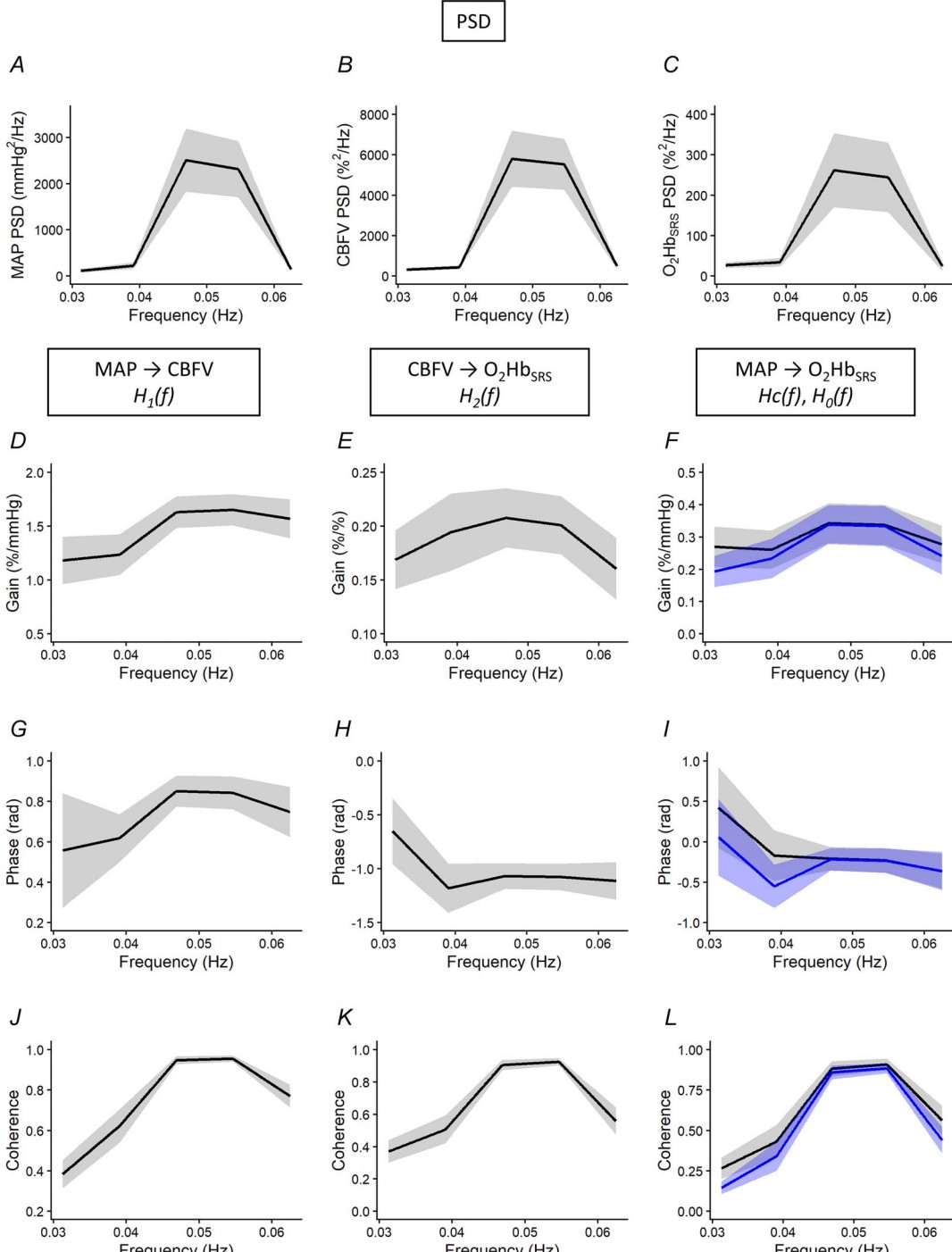

**Figure 3. Group-averaged power spectral density and transfer function metrics during repeated sit–stand manoeuvres at 0.05 Hz**

$A$–$C$, PSD of MAP, PSD and $O_2Hb_{SRS}$, respectively, showing the dominant 0.05 Hz oscillation produced by the sit–stand paradigm. $D$–$F$, gain spectra for the transfer functions between MAP→CBFV ($G_1$), CBFV→$O_2Hb_{SRS}$ ($G_2$) and for the cascade model ($G_c$) and empirically measured total function MAP→$O_2Hb_{SRS}$ ($G_0$). $G$–$I$, corresponding phase spectra for MAP→CBFV ($P_1$), CBFV→$O_2Hb_{SRS}$ ($P_2$), and for the cascade ($P_c$) and total MAP→$O_2Hb_{SRS}$ ($P_0$). $J$–$L$, coherence functions for MAP→CBFV ($C_1$), CBFV→$O_2Hb_{SRS}$ ($C_2$), and for the cascade ($C_c$) and total MAP→$O_2Hb_{SRS}$ ($C_0$). Thick lines denote the group means; grey shaded areas represent 95% confidence intervals across subjects. The cascade model ($H_1(f) \times H_2(f)$) is shown in blue. CBFV, cerebral blood flow velocity; MAP, mean arterial pressure; $O_2Hb_{SRS}$, oxyhaemoglobin calculated by spatially resolved spectroscopy method; PSD, power spectral density.

**Table 2. Power spectral density and transfer functional indices across frequency bands during rest**

| | Frequency | All | Women | Men | *P* |
|---|---|---|---|---|---|
| **PSD** | | | | | |
| MAP (mmHg$^2$/Hz) | VLF | 143.4 (116.7) | 115.1 (52.7) | 187.7 (168.6) | 0.051 |
| | LF | 23.7 (18.0) | 21.3 (11.6) | 27.6 (25.0) | 0.282 |
| | HF | 2.5 (5.7) | 1.8 (1.8) | 3.7 (9.0) | 0.852 |
| CBFV (%$^2$/Hz) | VLF | 203.4 (147.6) | 484.4 (410.1) | 767.7 (478.5) | 0.238 |
| | LF | 26.4 (13.8) | 69.6 (30.4) | 76.4 (43.7) | 0.235 |
| | HF | 2.5 (1.7) | 7.7 (7.5) | 6.6 (5.7) | 0.159 |
| O$_2$Hb$_{SRS}$ (%$^2$/Hz) | VLF | 39.0 (51.8) | 43.1 (61.4) | 32.7 (32.7) | 0.536 |
| | LF | 5.3 (8.6) | 4.1 (5.2) | 7.1 (12.1) | 0.281 |
| | HF | 0.9 (2.1) | 0.4 (0.4) | 1.8 (3.3) | 0.034 |
| **MAP–CBFV, $H_1(f)$** | | | | | |
| Gain ($G_1$) (%/mmHg) | VLF | 1.38 (0.72) | 1.24 (0.53) | 1.6 (0.93) | 0.118 |
| | LF | 1.56 (0.54) | 1.62 (0.46) | 1.46 (0.64) | 0.340 |
| | HF | 1.62 (0.73) | 1.70 (0.73) | 1.49 (0.73) | 0.372 |
| Phase ($P_1$) (radian) | VLF | 1.02 (0.38) | 0.98 (0.37) | 1.07 (0.41) | 0.515 |
| | LF | 0.61 (0.25) | 0.62 (0.28) | 0.61 (0.2) | 0.897 |
| | HF | 0.02 (0.26) | 0.04 (0.26) | −0.01 (0.27) | 0.613 |
| Coherence ($C_1$) | VLF | 0.44 (0.14) | 0.44 (0.14) | 0.45 (0.14) | 0.762 |
| | LF | 0.61 (0.17) | 0.64 (0.16) | 0.55 (0.17) | 0.096 |
| | HF | 0.48 (0.13) | 0.49 (0.13) | 0.46 (0.14) | 0.577 |
| **CBFV–O$_2$Hb$_{SRS}$, $H_2(f)$** | | | | | |
| Gain ($G_2$) (%/%) | VLF | 0.17 (0.08) | 0.19 (0.08) | 0.14 (0.08) | 0.057 |
| | LF | 0.17 (0.11) | 0.15 (0.09) | 0.19 (0.15) | 0.328 |
| | HF | 0.21 (0.23) | 0.15 (0.12) | 0.31 (0.32) | 0.027 |
| Phase ($P_2$) (radian) | VLF | −0.63 (0.5) | −0.59 (0.49) | −0.7 (0.27) | 0.768 |
| | LF | −0.96 (0.71) | −1.38 (0.45) | −0.71 (0.58) | 0.078 |
| | HF | −0.30 (0.48) | −0.22 (0.46) | −0.15 (0.48) | 0.108 |
| Coherence ($C_2$) | VLF | 0.47 (0.17) | 0.52 (0.16) | 0.45 (0.20) | 0.422 |
| | LF | 0.40 (0.14) | 0.48 (0.13) | 0.34 (0.12) | 0.009 |
| | HF | 0.29 (0.08) | 0.29 (0.08) | 0.25 (0.06) | 0.122 |
| **MAP–O$_2$Hb$_{SRS}$, $H_0(f)$** | | | | | |
| Gain ($G_0$) (%/mmHg) | VLF | 0.29 (0.16) | 0.32 (0.18) | 0.24 (0.09) | 0.102 |
| | LF | 0.30 (0.18) | 0.28 (0.14) | 0.33 (0.23) | 0.353 |
| | HF | 0.45 (0.45) | 0.31 (0.17) | 0.67 (0.64) | 0.011 |
| Phase ($P_0$) (radian) | VLF | 0.02 (0.59) | −0.02 (0.64) | −0.10 (0.50) | 0.496 |
| | LF | −0.42 (0.53) | −0.55 (0.50) | −0.21 (0.52) | 0.045 |
| | HF | −0.26 (0.50) | −0.41 (0.53) | −0.03 (0.34) | 0.016 |
| Coherence ($C_0$) | VLF | 0.36 (0.13) | 0.39 (0.13) | 0.32 (0.11) | 0.084 |
| | LF | 0.36 (0.12) | 0.39 (0.12) | 0.32 (0.11) | 0.041 |
| | HF | 0.28 (0.07) | 0.28 (0.08) | 0.26 (0.05) | 0.442 |

Values are means (SD). *P*-values refer to comparisons between men and women. VLF, very low frequency (0.02–0.07 Hz); LF, low frequency (0.07–0.20 Hz); HF, high frequency (0.20–0.35 Hz). CBFV, cerebral blood flow velocity; MAP, mean arterial pressure; O$_2$Hb$_{SRS}$, oxyhaemoglobin calculated by spatially resolved spectroscopy method; PSD, power spectral density;

critical coherence threshold (Table 3). The total coherence ($C_0$) and cascade coherence ($C_c$) overlapped at 0.05 Hz (Fig. 3*L*).

**Linear correlations between the total and cascade model transfer function estimates.** All transfer function indices (gain, phase and coherence) showed statistically significant correlations between the directly estimated ($G_0$, $P_0$ and $C_0$) and the cascade model metrics ($G_c$, $P_c$ and $C_c$) across all frequency bands, during both rest and sit–stand conditions (Table 5). During rest, the strongest correlations appeared in the LF range: gain ($G_0$ *vs.* $G_c$, $R^2 = 0.818$; Fig. 4*A*), phase ($P_0$ *vs.* $P_c$, $R^2 = 0.557$; Fig. 4*C*) and coherence ($C_0$ *vs.* $C_c$, $R^2 = 0.744$; Fig. 4*E*). During the sit–stand manoeuvres, statistically significant correlations were observed at 0.05 Hz: gain ($R^2 = 0.984$, Fig. 4*B*), phase ($R^2 = 0.993$, Fig. 4*D*) and coherence ($R^2 = 0.695$, Fig. 4*F*).

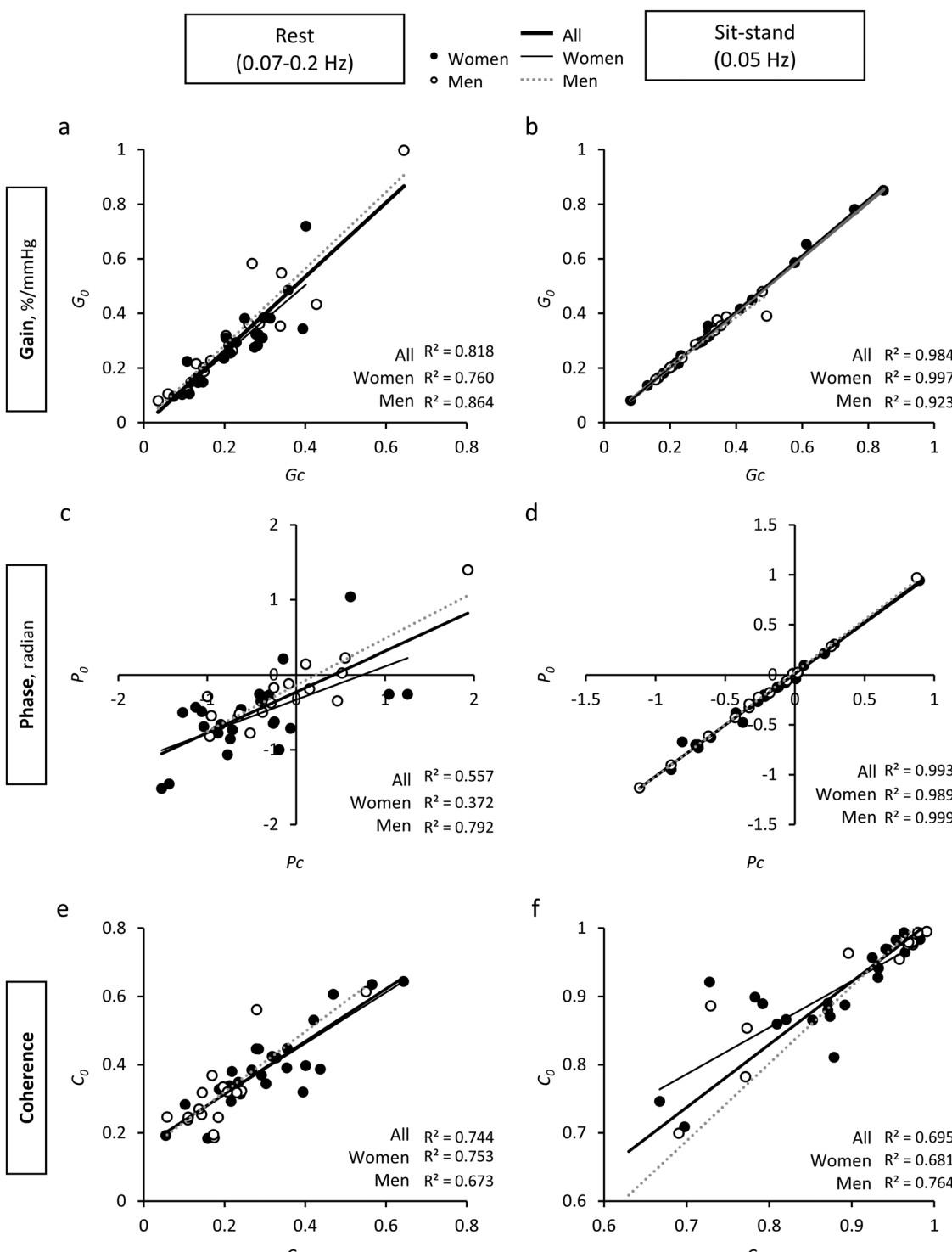

**Figure 4. Correlations between cascade model estimates ($H_c(f)$) and empirically derived total transfer function ($H_0(f)$) during rest (left panels) and sit–stand at 0.05 Hz (right panels)**

Each subplot shows linear regressions between corresponding parameters of $H_0$ and $H_c$: *A* and *B*, gain ($G_0$ *vs.* $G_c$); *C* and *D*, phase ($P_0$ *vs.* $P_c$); and *E* and *F*, coherence ($C_0$ *vs.* $C_c$). Resting-state metrics were computed within the low frequency range (0.07–0.2 Hz), while sit–stand values correspond to the driven 0.05 Hz oscillations. Filled circles represent data from women; open circles represent data from men. Thick lines denote regression across all participants, thin solid lines for women and dotted lines for men.

**Table 3. Number and percentage of subjects exceeding the 95% confidence threshold for each component of the cascade model during rest and sit–stand manoeuvres**

|  | Frequency | n (%) |
|---|---|---|
| MAP–CBFV, $H_1(f)$ |  |  |
|  | VLF | 31 (75.6) |
| Rest |  |  |
|  | LF | 38 (92.7) |
|  | HF | 36 (87.8) |
| Sit–stand | 0.05 Hz | 36 (100) |
| CBFV–O$_2$Hb$_{SRS}$, $H_2(f)$ |  |  |
|  | VLF | 30 (73.1) |
| Rest |  |  |
|  | LF | 26 (63.4) |
|  | HF | 15 (36.6) |
| Sit–stand | 0.05 Hz | 36 (100) |
| MAP–O$_2$Hb$_{SRS}$, $H_0(f)$ |  |  |
|  | VLF | 20 (48.8) |
| Rest |  |  |
|  | LF | 20 (48.8) |
|  | HF | 13 (31.7) |
| Sit–stand | 0.05 Hz | 36 (100) |

VLF, very low frequency (0.02–0.07 Hz); LF, low frequency (0.07–0.20 Hz); HF, high frequency (0.20–0.35 Hz).

**Detection of significant coupling across MAP–CBFV–O$_2$Hb$_{SRS}$ pathways.** During rest, significant coupling was observed in 90% of subjects for $H_1$, 61% for $H_2$ and 42% for $H_0$, with the highest detection rate occurring in the LF band of $H_1$ (92%). During sit–stand manoeuvres, the proportion of subjects exhibiting significant coupling increased markedly, reaching 100% for $H_1$, $H_2$ and $H_0$ at the forced oscillation frequency of 0.05 Hz (Table 3; see also Appendix Fig. A3).

## Discussion

This study investigated the relationship between the upstream dCA and downstream MF using a two-component cascade model in young healthy volunteers. We found strong correlations between the cascade model derived transfer function indices ($G_c$, $P_c$, $C_c$) and the directly estimated total transfer function indices ($G_0$, $P_0$, $C_0$) during both rest and sit–stand manoeuvres at 0.05 Hz. Notably, large BP, CBFV and O$_2$Hb$_{SRS}$ oscillations during sit–stand manoeuvres markedly enhanced the correlation between the cascade model derived and the directly estimated total transfer function indices. These findings support the concept that dCA and MF operate serially to regulate how systemic blood pressure fluctuations propagate from large arteries to the cortical microcirculation to influence brain perfusion and tissue oxygenation, with their combined effects accounting for the integrated BP–O$_2$Hb relationship.

**Insights from gain.** Transfer function gain from MAP to CBFV ($G_1$) was relatively high, whereas CBFV to O$_2$Hb$_{SRS}$ gain ($G_2$) was consistently lower, indicating that attenuation of blood pressure oscillatory transmissions to brain perfusion occurs mainly at the microvascular level. This attenuation, observed both at rest and during sit–stand manoeuvres, aligns with prior work showing smaller O$_2$Hb responses to changes in blood pressure compared with that of CBFV (Mol et al., 2021; Tarumi et al., 2014; van Beek et al., 2012). These findings are further consistent with evidence that small cerebral blood vessels contribute nearly half of the total cerebrovascular resistance, with arterioles and capillaries accounting for the majority of resistance in the cortical circulation (De Silva & Faraci, 2016; Faraci, 2011; Gould et al., 2017; Iadecola, 2017).

The cascade model gain ($G_c$) during rest under spontaneous oscillatory conditions mirrored that of directly estimated $G_0$ (Fig. 2*F*), but was lower than $G_0$ at higher frequencies above 0.2 Hz. This discrepancy may reflect the fact that both the amplitude of blood pressure and CBFV oscillations were small at these higher frequencies (as revealed by the power spectral analysis), and thus a low signal/noise ratio may have led to an underestimated cascade gain. It is also possible that non-linear blood flow regulation of the microcirculation as revealed by a low coherence (presented in Fig. 2*K*) at these higher frequencies biased the linear cascade model estimation of $G_c$. With large oscillations during sit–stand manoeuvres at 0.05 Hz – an improved signal/noise ratio – $G_c$ converged with $G_0$, associated with increases in coherence function (Fig. 3), consistent with the linear cascade model hypothesis.

**Insights from Phase.** The phase between MAP to CBFV ($P_1$) was positive during rest in the VLF and LF ranges and during sit–stand manoeuvres at 0.05 Hz and decreased gradually with increases in frequency. These observations are consistent with prior studies suggesting that cerebral autoregulatory effects cause CBFV fluctuations to shift toward a phase lead relative to changes in blood pressure (Claassen et al., 2016; van Beek et al., 2012; Zhang et al., 1998). In contrast, the phase between CBFV and O$_2$Hb$_{SRS}$ ($P_2$) was negative during rest in the VLF and LF ranges and during sit–stand manoeuvres at 0.05 Hz which reflects the delayed microvascular responses to changes in upstream blood flow oscillations. Similar negative phases have been reported between CBFV and tissue oxygenation (Mol et al., 2021; Tarumi et al., 2014; van Beek et al., 2012). The cascade model $P_c$ showed similar patterns to the directly estimated total $P_0$ in that both approach zero at the VLF

**Table 4. Power spectral density and transfer functional indices at 0.05 Hz during sit–stand manoeuvres**

| | All | Women | Men | *P* |
|---|---|---|---|---|
| PSD | | | | |
| MAP (mmHg$^2$/Hz) | 2410.2 (2098.9) | 2259.9 (1891.3) | 2676.0 (2484.2) | 0.575 |
| CBFV (%$^2$/Hz) | 5661.4 (4281.5) | 4552.6 (3091.1) | 7623.3 (5424.4) | 0.126 |
| O$_2$Hb$_{SRS}$ (%$^2$/Hz) | 252.9 (286.5) | 285.4 (343.0) | 195.3 (134.4) | 0.780 |
| MAP–CBFV, $H_1(f)$ | | | | |
| Gain ($G_1$) (%/mmHg) | 1.64 (0.47) | 1.52 (0.41) | 1.85 (0.52) | 0.040 |
| Phase ($P_1$) (radian) | 0.85 (0.26) | 0.88 (0.24) | 0.79 (0.28) | 0.301 |
| Coherence ($C_1$) | 0.95 (0.06) | 0.95 (0.05) | 0.95 (0.07) | 0.958 |
| CBFV–O$_2$Hb$_{SRS}$, $H_2(f)$ | | | | |
| Gain ($G_2$) (%/%) | 0.20 (0.09) | 0.23 (0.10) | 0.17 (0.06) | 0.045 |
| Phase ($P_2$) (radian) | −1.07 (0.39) | −1.10 (0.35) | −1.02 (0.47) | 0.571 |
| Coherence ($C_2$) | 0.91 (0.08) | 0.92 (0.07) | 0.91 (0.10) | 0.918 |
| MAP–O$_2$Hb$_{SRS}$, $H_0(f)$ | | | | |
| Gain ($G_0$) (%/mmHg) | 0.34 (0.19) | 0.36 (0.21) | 0.30 (0.13) | 0.410 |
| Phase ($P_0$) (radian) | −0.22 (0.45) | −0.22 (0.42) | −0.22 (0.53) | 0.999 |
| Coherence ($C_0$) | 0.89 (0.12) | 0.90 (0.08) | 0.88 (0.17) | 0.576 |

Values are mean (SD). *P*-values refer to comparisons between men and women. CBFV, cerebral blood flow velocity; MAP, mean arterial pressure; O$_2$Hb$_{SRS}$, oxyhaemoglobin calculated by spatially resolved spectroscopy method; PSD, power spectral density.

**Table 5. Pearson correlation coefficients between total transfer function ($H_0(f)$) and cascade model ($H_c(f)$) transfer function across frequency bands during rest and during sit–stand manoeuvres**

| | Frequency | Pearson's *r* (95% CI) | *P* |
|---|---|---|---|
| Gain ($G_0$–$G_c$) (%/mmHg) | | | |
| | VLF | 0.86 (0.77–0.95) | <0.001 |
| Rest | | | |
| | LF | 0.91 (0.80–0.96) | <0.001 |
| | HF | 0.94 (0.93–0.98) | <0.001 |
| Sit–stand | 0.05 Hz | 0.99 (0.97–1.00) | <0.001 |
| Phase ($P_0$–$P_c$) (radian) | | | |
| | VLF | 0.61 (0.41–0.76) | <0.001 |
| Rest | | | |
| | LF | 0.75 (0.54–0.88) | <0.001 |
| | HF | 0.50 (0.21–0.72) | 0.001 |
| Sit–stand | 0.05 Hz | 1.00 (0.99–1.00) | <0.001 |
| Coherence ($C_0$–$C_c$) | | | |
| | VLF | 0.70 (0.47–0.84) | <0.001 |
| Rest | | | |
| | LF | 0.86 (0.73–0.94) | <0.001 |
| | HF | 0.78 (0.61–0.88) | <0.001 |
| Sit–stand | 0.05 Hz | 0.83 (0.74–0.94) | <0.001 |

The *r* and *P*-values were calculated using Pearson's correlation coefficients between $H_0(f)$ and $H_c(f)$. VLF, very low frequency (0.02–0.07 Hz); LF, low frequency (0.07–0.20 Hz); HF, high frequency (0.20–0.35 Hz).

and HF frequencies and were negative in the LF band (Fig. 2*I*). The statistically significant linear correlations between $P_c$ and $P_0$ further support the cascade model as a physiologically grounded representation of how dCA and MF jointly regulate the temporal propagation of BP and CBF oscillations from the macro- to microcirculation.

**Insights from coherence.** The group-averaged magnitude-squared coherence between MAP and CBFV ($C_1$) was ~0.6 in the LF range, indicating a sufficiently strong linear association to support reliable estimation of gain and phase in $H_1$. In contrast, lower coherence in the VLF range likely reflects increased nonlinearity in

the MAP–CBFV relationship (Zhang et al., 1998). The coherence between CBFV and $O_2Hb_{SRS}$ ($C_2$) was ∼0.5 in the VLF range, but declined progressively across the LF and HF bands, suggesting weakening linear coupling in $H_2$ with increasing frequency during spontaneous fluctuations. At rest, the cascade model coherence ($C_c$) followed a pattern similar to $C_0$, but showed lower values at frequencies above 0.2 Hz. This observation is expected from multiplying component coherences with the cascade formulation. During sit–stand manoeuvres at 0.05 Hz, both $C_0$ and $C_c$ approached to 1, indicating highly reliable linear coupling under forced oscillations.

**State-dependent coupling in the cascade model.** State-dependent coupling analysis supports the proposed cascade structure. During spontaneous fluctuations, the highest proportion of subjects exhibited significant coupling in $H_1$, particularly in the LF range, indicating robust pressure–flow transmission at the macrovascular level. In contrast, the proportion of subjects with significant coupling gradually decreased for $H_2$ and $H_0$, suggesting increasing in nonlinearity and attenuation of blood flow fluctuations toward the microcirculation. During sit–stand manoeuvres, however, significant coupling was observed in all subjects across $H_1$, $H_2$ and $H_0$, indicating robust engagement of the full MAP→CBFV→$O_2Hb$ cascade. These findings provide individual-level support for the cascade model and highlight the frequency- and state-dependence of cerebrovascular coupling.

**Sex differences.** Although sex-related variation in cerebrovascular physiology is well known, its impact on dCA and MF is less clear (Duque et al., 2017). During sit–stand manoeuvres, men had higher $G_1$ while women had higher $G_2$, possibly reflecting greater microvascular compliance in young women related to oestrogen-mediated vasodilation (Kastrup et al., 1998). At rest, women showed higher $G_2$ in the VLF band but lower in the HF band, with no sex differences in $G_1$. These findings suggest the presence of sex-related differences in cerebral macro- and microvascular regulation which may vary by frequency. However, the absence of controls for menstrual cycle phase in the present study limits interpretation. Notably, prior work indicates that menstrual cycle phases exert only a minor influence on dCA (Favre & Serrador, 2019).

**Comparisons with other studies.** Prior studies by Mol et al. (2021) and Elting et al. (2020) attributed the phase differences between BP→CBFV and BP→$O_2Hb$ to passive microvascular effects that introduce a NIRS signal delay, becoming more apparent in the HF range above 0.2 Hz (Elting et al., 2020; Mol et al., 2021). Correcting for this offset by subtracting the corresponding phase lag improved agreement between BP→CBFV and BP→$O_2Hb$. Consistent with these observations, we found a negative phase (i.e. time delay) between changes in CBFV and $O_2Hb_{SRS}$, both at rest in the VLF and LF ranges and during sit–stand manoeuvres at 0.05 Hz. A portion of this negative phase may indeed reflect a passive time delay in the transmission of upstream CBF oscillations into microvascular $O_2Hb_{SRS}$ changes (Fig. 2).

However, the passive-delay interpretation alone cannot fully account for observations from studies which measured microvascular flow directly with diffuse correlation spectroscopy (DCS) (Favilla et al., 2023; Shoemaker et al., 2023). For example, Favilla et al. reported that the BP–CBFV and BP–DCS phase differences observed in healthy controls were diminished in stroke patients, implying the loss of microvascular phase contributions following focal brain injury (Favilla et al., 2023). In addition, Shoemaker et al. (2023) showed that during rapid hypotension, CBF was preserved through a rapid increase in cerebral large artery compliance, whereas microvascular resistance adapted more slowly (Shoemaker et al., 2023). These observations suggest the presence of both frequency- and physiological state-dependent dissociations in the macro- and microvascular blood flow regulation, indicative of dynamic, compartment-specific coupling rather than a passive propagation delay (Favilla et al., 2023; Shoemaker et al., 2023). The cascade model presented here provides a physiological framework to capture the hierarchical organization of macro- and microvascular contributions to cerebral autoregulation.

**Methodological considerations and limitations.** Several methodological issues merit consideration. First, coherence estimation is used in linear transfer function analysis to assess the reliability of gain and phase estimates; however, its interpretation depends on the spectral estimation procedures employed, which include the data segment length and the type of data window used to minimize spectral leakage (Panerai et al., 2023). Consequently, significance thresholds are not universal and vary across analysis settings. To ensure comparability and statistical rigor, we adopted the standardized CARNet framework, which derives robust significance thresholds from Monte Carlo simulations of independent white Gaussian noise. Accordingly, significance was evaluated against the 95% confidence limit corresponding to use of Hanning windows with 50% overlap, yielding a critical coherence value of 0.34 based on five overlapping windows in our analysis (Panerai et al., 2023).

While coherence provides an important measure of reliability, it is inherently non-directional and cannot distinguish feedforward from feedback interactions. Recent work has shown that frequency-domain non-linear approaches incorporating

directionality may provide additional insight into cardiovascular–cerebrovascular interactions, particularly under conditions such as orthostatic stress where bidirectional coupling may be present (Porta et al., 2023). Integration of directionality metrics alongside coherence-based reliability assessment may therefore refine the application of the cascade model to understand the complexity of these interactions.

Second, the cascade model assumes linearity, which limits its ability to capture nonlinearity of blood flow regulation (Marmarelis et al., 2012; Mitsis et al., 2004). Although nonlinearity analyses were not performed in this study, the validity of linear TFA in the resting state is supported by prior evidence showing that the low coherence between BP and CBFV, often most prominent in the VLF band, may reflect the influence of unaccounted physiological inputs rather than the presence of intrinsic nonlinear dynamics (Panerai et al., 2006; Peng et al., 2008, 2010). In supporting this hypothesis, Porta et al. (2020) reported that even in a clinical population of patients undergoing aortic replacement surgery, linear approximations remained robust in the MAP–CBFV coupling (Porta et al., 2020). In the present study during sit–stand manoeuvres, where large oscillations of BP, CBFV and $O_2Hb_{SRS}$ were induced, the estimated transfer function indices and the cascade model estimates converged at 0.05 Hz, indicating that the linear cascade approximation captured the dominant dynamics under forced oscillatory conditions. More broadly, increased MAP–CBFV coupling during standing or repeated sit–stand manoeuvres has been consistently reported across linear transfer-function analyses, wavelet-based approaches and nonlinear directionality frameworks (Claassen et al., 2009; Porta et al., 2026; Smirl et al., 2015), suggesting that the primary pressure–flow relationships represented in the cascade model are robust across analytical approaches.

Third, the cascade model represents BP→CBFV→$O_2Hb$ dynamics as an open-loop series system. While some directionality- or causality-based analyses have suggested closed-loop characteristics within the BP–CBFV relationship (Bari et al., 2017; Saleem et al., 2018), it is yet unclear if these statistical interdependencies represent physiological feedback mechanisms or simply shared regulatory mechanisms between cerebrovascular–cardiovascular systems. Accordingly, the present framework focuses on feed-forward transmission of pressure fluctuation across the vascular tree. This approach is particularly suited to forced oscillatory conditions, where feedforward components are expected to dominate the observed dynamics (Porta et al., 2026). Future development of the cascade framework to include closed-loop formulations may be warranted to better elucidate the feed-forward and feedback dynamics across different clinical states (Gelpi et al., 2025; Schmidt et al., 2018).

Fourth, the PSDs of HR and $EtCO_2$ exhibited distributions similar to those of MAP, CBFV and $O_2Hb_{SRS}$ under both rest and sit–stand conditions. Since beat-to-beat changes in HR and breath-by-breath changes in $EtCO_2$ likely correlate with these signals, their potential influence on the cascade modelling of BP–CBFV–$O_2Hb$ dynamics cannot be excluded. The possible confounding effects of HR and $EtCO_2$ on the cascade relationships therefore warrants further studies.

Finally, several technical limitations may have influenced signal quality. Beat-to-beat MAP data occasionally has gaps due to Finapres calibration. Although linear interpolation of missing segments was performed as recommended, residual error cannot be fully excluded (Claassen et al., 2016; Panerai et al., 2023). TCD-based estimated changes in CBF assumed a constant MCA diameter, which may not hold under all conditions (Larsen et al., 1994). CBFV was measured unilaterally in the right MCA, precluding assessment of potential hemispheric asymmetries, although, bilateral concordance of MCA flow velocity has been demonstrated in healthy adults both at rest and during non-lateralizing physiological stimuli such as exercise (Billinger et al., 2017; Sorteberg et al., 1990; Weston et al., 2023; Yonan et al., 2014). Head motion may introduce baseline shifts in nTHI or TOI signal, which were corrected after visual inspection when presented. Moreover, even with SRS processing, NIRS signals include mixed arterial–venous contributions and extracranial contamination (Al-Rawi et al., 2001; Hogue et al., 2021; Terborg et al., 2003). For these reasons, we interpreted both TCD and NIRS measurements in terms of relative, rather than absolute, physiological values for TFA.

**Clinical implications.** TCD-CBFV and NIRS-derived indices of CBF regulation often diverge, suggesting modality-specific sensitivity to different vascular compartments. For instance, in healthy adults, Thudium et al. (2023) found poor agreement between TCD-CBFV and NIRS-derived dCA indices during haemodynamic challenges induced by incremental lower body negative or positive pressure (Thudium et al., 2023). Similarly, in subarachnoid haemorrhage, Budohoski et al. (2013) observed only moderate inter-modality correlations (Budohoski et al., 2013), while in cardiac arrest, Lim et al. (2024) reported discordant autoregulatory patterns between CBF velocity and cerebral oxygenation signals (Lim et al., 2024). Likewise, in comatose brain-injured patients, Rivera-Lara et al. (2017) found low-to-moderate correlations between TCD and NIRS assessments of autoregulatory status (Rivera-Lara et al., 2017). Finally, Muller et al., comparing MAP–CBFV with CBFV–$O_2Hb$ transfer functions, reported impaired dCA but preserved MF in treated chronic hypertension (Müller et al., 2020).

Collectively, these findings align with the findings of the present study indicating that TCD and NIRS

likely probe complementary, rather than interchangeable, components of cerebrovascular regulation. The cascade framework explicitly formalizes this compartmental separation as dynamic serial processes. Nevertheless, the present conclusions are derived from healthy participants and should be interpreted within this context. Whether the cascade model generalizes to ageing or pathological states remains to be determined and warrants targeted investigation in populations with altered vascular or metabolic control.

Beyond NIRS and TCD, the cascade model may also inform interpretation of fMRI blood oxygen level-dependent (BOLD) signals, where low frequency blood pressure oscillations have been shown to influence BOLD dynamics, but are often treated as physiological 'noise' rather than markers of cerebrovascular regulation (Golestani et al., 2015; Whittaker et al., 2019). Finally, in future applications, cascade-derived transfer function indices may also serve as endpoints in clinical trials aimed at restoring compartment-specific cerebrovascular regulation.

**Conclusion.** This study demonstrates that dynamic cerebrovascular regulation can be represented as a cascade model of the upstream dCA and downstream MF. In young healthy volunteers, this two-component model captured the integrated BP–$O_2$Hb relationship, with higher fidelity under forced haemodynamic oscillations during sit–stand manoeuvres. These findings establish a conceptual and methodological framework for dissecting compartment-specific contributions to cerebrovascular control and provide a translational tool for future mechanistic and clinical applications.

## Appendix

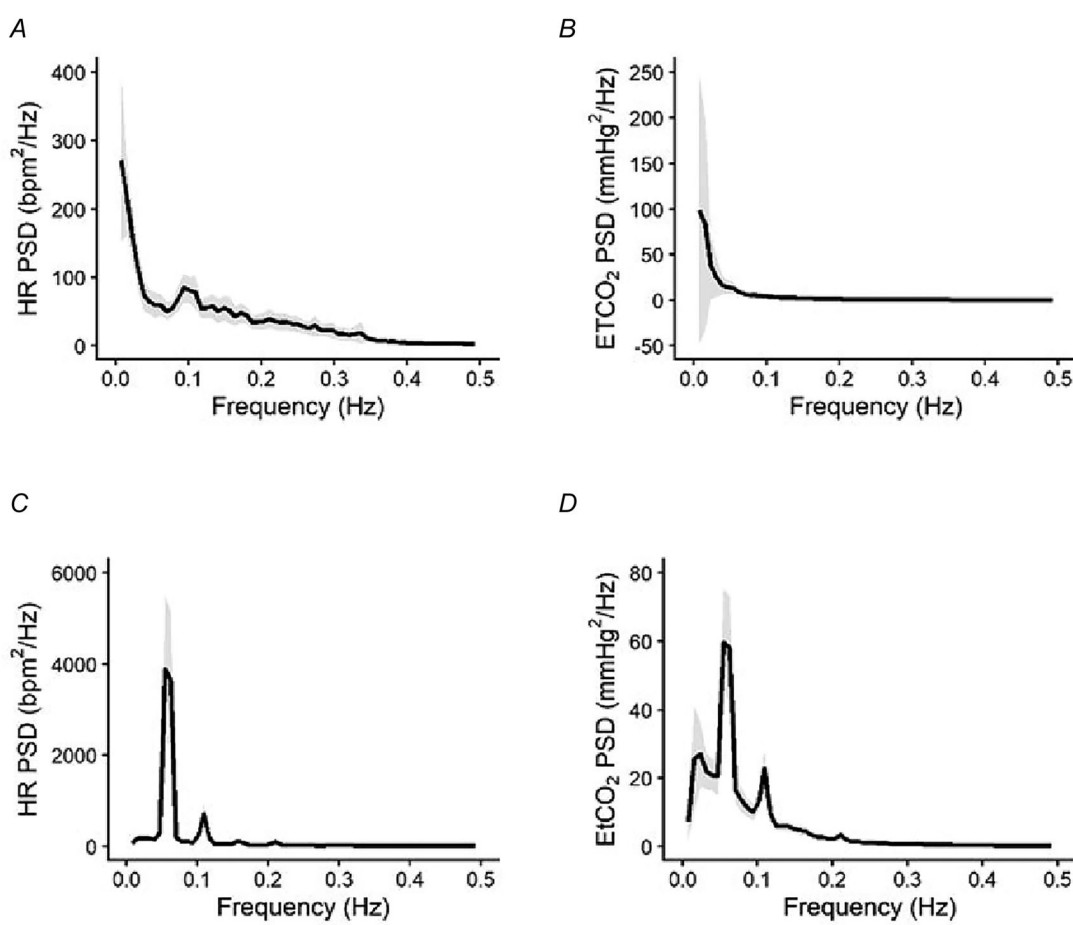

**Figure A1. Group-averaged power spectral densities for HR and EtCO2 during rest and repeated sit-stand manoeuvres**

*A* and *B*, PSD of HR and EtCO$_2$ during the resting period. *C* and *D*, PSD of HR and EtCO$_2$ during repeated sit–stand manoeuvres. Thick black lines denote the group mean and grey shaded areas represent the 95% confidence intervals. EtCO$_2$, end-tidal carbon dioxide; HR, heart rate; PSD, power spectral density.

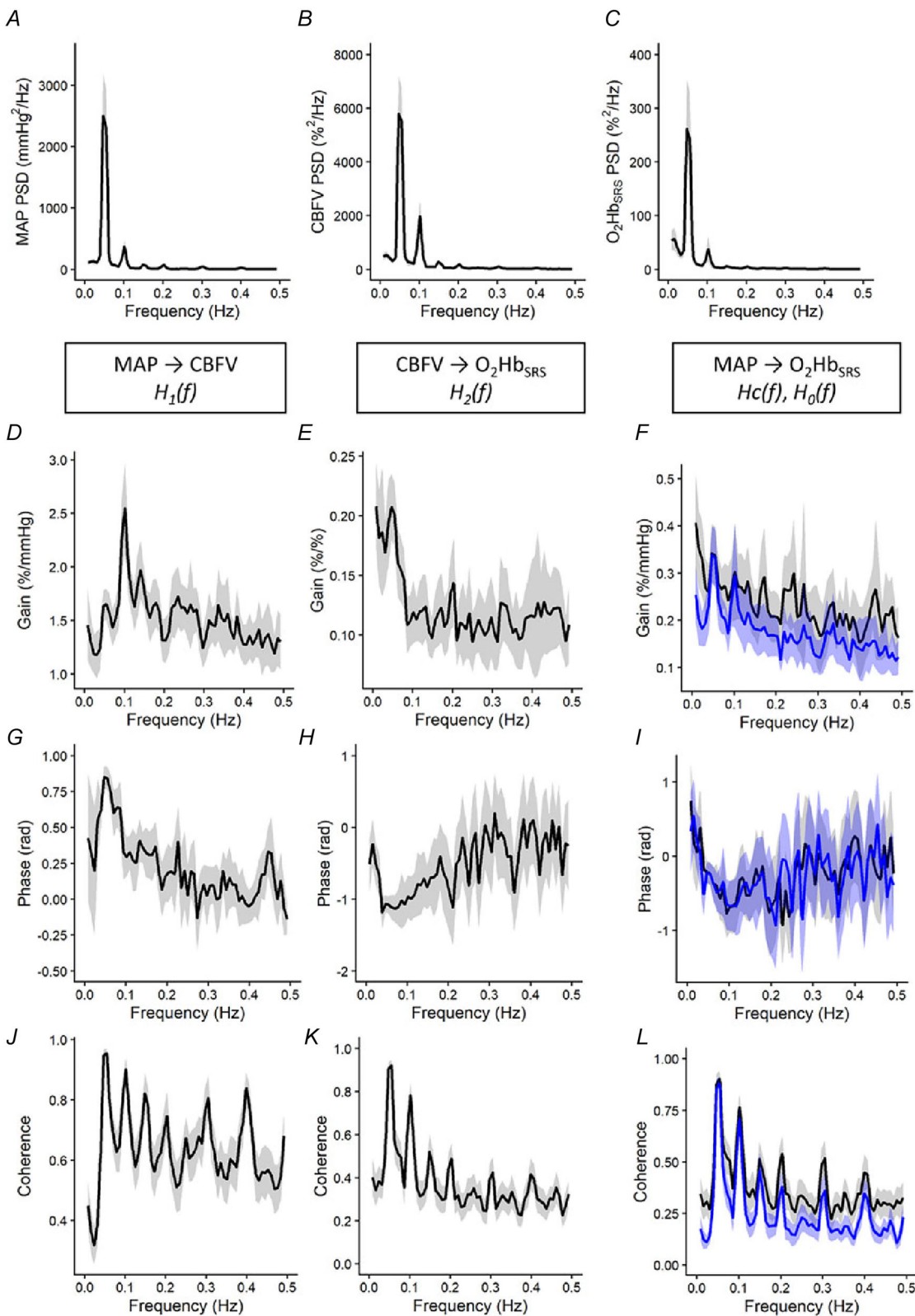

**Figure A2. Group-averaged power spectral density and transfer function metrics during repeated sit-stand manoeuvres across the full frequency range (0–0.5 Hz)**
*A–C*, PSD of MAP, CBFV and $O_2Hb_{SRS}$. *D–F*, gain spectra for transfer functions MAP→CBFV ($G_1$), CBFV→$O_2Hb_{SRS}$ ($G_2$), the modelled cascade ($G_c = G_1 \times G_2$) and empirically measured total MAP→$O_2Hb_{SRS}$ relationship ($G_0$). *G–I*, corresponding phase spectra for MAP→CBFV ($P_1$), CBFV→$O_2Hb_{SRS}$ ($P_2$), the cascade model ($P_c$) and total

MAP$\rightarrow$O$_2$Hb$_{SRS}$($P_0$). *J–L*, coherence functions for MAP$\rightarrow$CBFV ($C_1$); CBFV$\rightarrow$O$_2$Hb$_{SRS}$ ($C_2$) and the cascade ($C_c$) and total MAP$\rightarrow$O$_2$Hb$_{SRS}$ ($C_0$). Thick black lines denote group means and grey shaded areas represent the 95% confidence intervals across subjects. The cascade model ($H_1(f) \times H_2(f)$) is shown in blue. CBFV, cerebral blood flow velocity; MAP, mean arterial pressure; O$_2$Hb$_{SRS}$, oxyhaemoglobin calculated by the spatially resolved spectroscopy method; PSD, power spectral density.

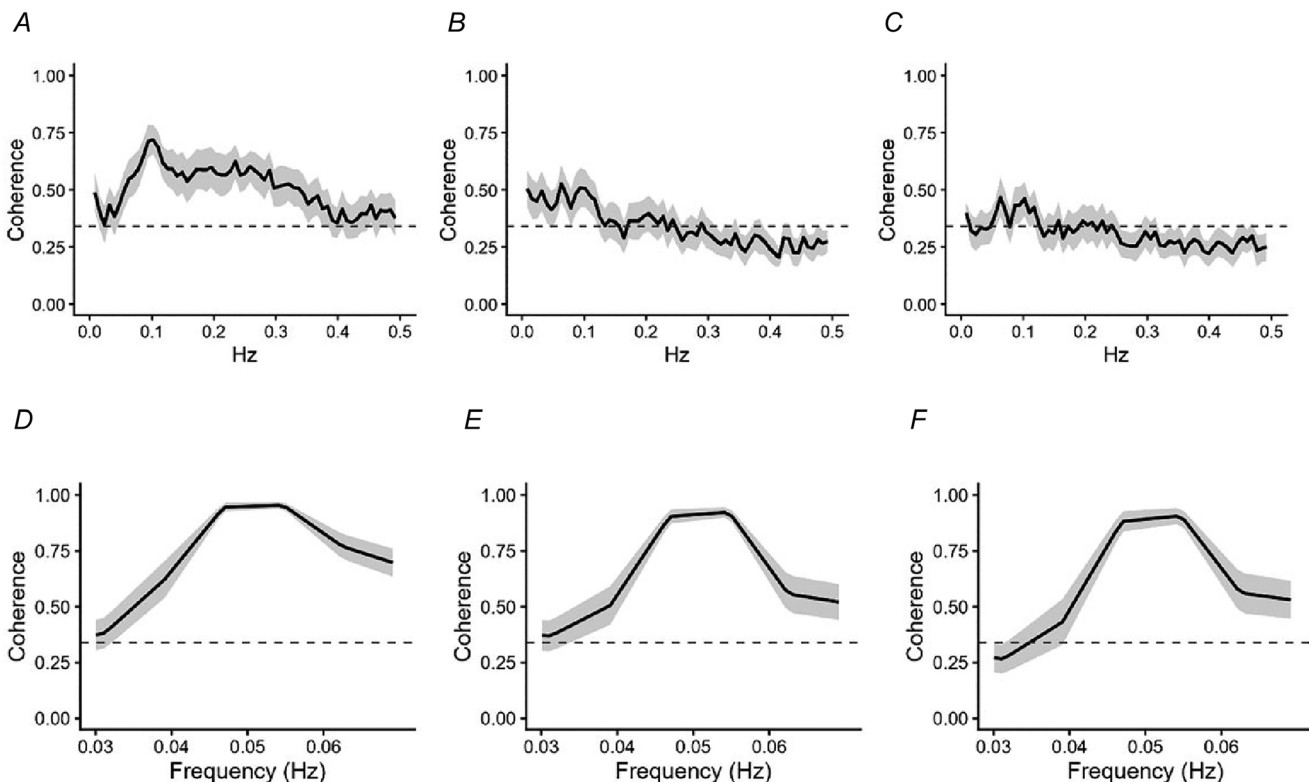

**Figure A3. Group-averaged coherence with indication of critical coherence during rest and sit-stand manoeuvres**

*A–C*, coherence spectra for MAP$\rightarrow$CBFV ($C_1$), CBFV$\rightarrow$O$_2$Hb$_{SRS}$ ($C_2$) and MAP$\rightarrow$O$_2$Hb$_{SRS}$ ($C_0$) during rest period. *D–F*, coherence spectra for MAP$\rightarrow$CBFV ($C_1$), CBFV$\rightarrow$O$_2$Hb$_{SRS}$ ($C_2$) and MAP$\rightarrow$O$_2$Hb$_{SRS}$ ($C_0$) during sit–stand manoeuvres. Thick lines denote group means and the grey shading represents 95% confidence intervals across subjects. Thin dotted lines indicate the critical coherence threshold (0.34), corresponding to the 95% significance limit for five overlapping spectral windows, following recommendations of the updated CARNet white paper (Panerai et al., 2023).

**Table A1. Number and proportion of subjects exhibiting negative phase values at each frequency (<0.10 Hz) during the resting phase**

| Observation | Frequency (Hz) | | | | | | | | | | | |
|---|---|---|---|---|---|---|---|---|---|---|---|---|
| | 0.0078 | 0.0156 | 0.0234 | 0.0313 | 0.0391 | 0.0469 | 0.0547 | 0.0625 | 0.0703 | 0.0781 | 0.0859 | 0.0938 |
| *n* | 17 | 15 | 15 | 16 | 10 | 4 | 5 | 0 | 3 | 1 | 1 | 0 |
| % | 41 | 36 | 36 | 39 | 24 | 9 | 12 | 0 | 7 | 2 | 2 | 0 |

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

## Additional information

### Data availability statement

The data that support the findings of this study are available from the corresponding author upon reasonable request.

### Competing interests

The authors declare no conflicts of interest.

### Author contributions

T.K. and R.Z. conceptualized and designed the study; D.C., T.T., J.W. and J.A. performed data acquisition; T.K., T.T., J.W. and R.W. performed data analysis. K.S., T.K. and J.C. interpreted the results, K.S. and T.K. drafted the manuscript. All authors reviewed and critically edited the manuscript for important intellectual content. All authors have read and approved the final version of this manuscript and agree to be accountable for all aspects of the work in ensuring that questions related to the accuracy or integrity of any part of the work are appropriately investigated and resolved. All persons designated as authors qualify for authorship, and all those who qualify for authorship are listed.

### Funding

This work was supported by NIH NIA R01AG033106 and NIH NHLBI R01HL102457. The content is solely the responsbility of the authors and does not necessarily represent the views of the National Institutes of Health.

### Acknowledgements

The authors thank all our study participants for their willingness, time and effort devoted to this study.

## Keywords

cascade model, cerebral autoregulation, microvascular function, near infrared spectroscopy, transfer function analysis

## Supporting information

Additional supporting information can be found online in the Supporting Information section at the end of the HTML view of the article. Supporting information files available:

**Peer Review History**

