## [Peer Review History · The Journal of Physiology]

A Cascade Model of Dynamic Cerebral Autoregulation

Takuya Kurazumi, Kartavya Sharma, Ricardo Wennekers, Tsubasa Tomoto, Danilo Cardim, Junyeon Won, John Ashley, Jurgen A.H.R. Claassen, and Rong Zhang

DOI: 10.1113/JP290519

Corresponding author(s): Rong Zhang (rongzhang@texashealth.org)

The following individual(s) involved in review of this submission have agreed to reveal their identity: Agnieszka Uryga (Referee #1)

Review Timeline:	Submission Date:	14-Nov-2025
	Editorial Decision:	12-Jan-2026
	Revision Received:	21-Mar-2026
	Accepted:	07-Apr-2026

Senior Editor: Eleonora Grandi

Reviewing Editor: Diego Candia-Rivera

Transaction Report:

Re: JP-RP-2025-290519 "A Cascade Model of Dynamic Cerebral Autoregulation" by Takuya Kurazumi, Kartavya Sharma, Ricardo Wennekers, Tsubasa Tomoto, Danilo Cardim, Junyeon Won, John Ashley, Jurgen A.H.R. Claassen, and Rong Zhang

Dear Dr Zhang,

Thank you for submitting your manuscript to The Journal of Physiology. It has been assessed by a Reviewing Editor and by 2 expert referees and we are pleased to tell you that it is potentially acceptable for publication following satisfactory major revision.

Please address all the points raised and incorporate all requested revisions or explain in your Response to Referees why a change has not been made. We hope you will find the comments helpful and that you will be able to return your revised manuscript within 2 months. If your article is NOT for a Special Issue, you may have 9 months to revise. If you require an extension, please contact journal staff: jp@physoc.org. Please note that this letter does not constitute a guarantee for acceptance of your revised manuscript.

REVISION CHECKLIST:

We look forward to receiving your revised submission.

Yours sincerely,

Eleonora Grandi
Senior Editor
The Journal of Physiology

REQUIRED ITEMS

- The Journal of Physiology funds authors of provisionally accepted papers to use the premium BioRender site to create high resolution schematic figures. Follow this link and enter your details and the manuscript number to create and download figures. Upload these as the figure files for your revised submission. If you choose not to take up this offer, we require figures to be of similar quality and resolution. If you are opting out of this service to authors, state this in the Comments section on the Detailed Information page of the submission form. The link provided should only be used for the purposes of this submission. Authors will be charged for figures created on this premium BioRender account if they are not related to this manuscript submission.

- A Data Availability Statement is required for all papers reporting original data. This must be in the Additional Information section of the manuscript itself. It must have the paragraph heading 'Data Availability Statement'. All data supporting the results in the paper must be either: in the paper itself; uploaded as Supporting Information for Online Publication; or archived in an appropriate public repository. The statement needs to describe the availability or the absence of shared data. Authors must include in their statement: a link to the repository they have used, or a statement that it is available as Supporting Information; reference the data in the appropriate sections(s) of their manuscript; and cite the data they have shared in the References section. Whenever possible, the scripts and other artefacts used to generate the analyses presented in the paper should also be publicly archived. If sharing data compromises ethical standards or legal requirements then authors are not expected to share it, but must note this in their statement. For more information, see our Statistics Policy.

EDITOR COMMENTS

Reviewing Editor:

Ethics Concerns:
Ethics approved, number missing

Comments to the Author:

The manuscript has now been reviewed by 2 expert referees. In view of referees comments, your manuscript would not be sufficient for publication in its current form. Please consider submitting a revised version for further consideration.

Senior Editor:

Comments for Authors to ensure the paper complies with the Statistics Policy:
Please ensure the paper complies with the Journal of Physiology's Statistics Policy.

Comments to the Author:

I concur with the Reviewing Editor's recommendation. Please carefully revise the manuscript to address all the reviewer comments to clarify and discuss several of the modeling aspects and deepen the analysis.

REFEREE COMMENTS

Referee #1:

Please add a rationale for using retrospective data instead of prospective data. There is no information on patient recruitment, and the main text does not provide the number of the Bioethical Committee approval.

"High temporal resolution" - could you provide the exact value?

Why was only unilateral TCD used? Shouldn't the choice depend on the dominant brain hemisphere? What was the sampling frequency?

Which type of Finapres did you use-Finapres NOVA or Finapres MIDI?

What does this mean: "with the NIRO device providing external output at 5 Hz"?

Please provide a reference here: "Participants were instructed to refrain from high-intensity exercise, caffeine, and alcohol for at least 24 hours prior to testing."

Did you control only EtCO₂, or tidal volume as well?

How were data averaged across the cardiac cycle? Please describe this sentence in more detail: "Beat-to-beat values of MAP, CBFV, TOI, and nTHI were analyzed using Acqknowledge software (BIOPAC Systems, Goleta, CA, USA) and averaged across cardiac cycles (Zhang et al., 1998)."

Why is there a '#' symbol before (1) in the formula?

Please add a reference to the latest CARNET paper in the "Transfer function analysis" section.

It should probably be "cross-spectrum" instead of "crossspectrum."

"The phase describes the temporal offset between the input and output (reflecting the time constant and/or delay of the vascular adjustment in response to changes in blood pressure or blood flow). The coherence represents the consistency of a linear relationship between the input and output across frequencies." - This is correct, but a general interpretation of phase shift and gain could also be helpful here.

In all formulas (3a, 3b, 3c), the names of Spv, Svc, etc., should be provided.

"To avoid 'phase wrapping', negative phase values below 0.10 Hz during rest were excluded from the phase calculation." - Please comment on how often such values were observed.

"0.05 Hz during repeated sit-stand maneuvers." - Is there any range around this frequency? Did you check whether there were fluctuations in the spectrum?

I suggest using either mean {plus minus} SD with parametric tests or median {plus minus} IQR with non-parametric tests. Mixing the two approaches results in different statistical power.

"Men had significantly greater height, weight, and BMI than women." - This finding is rather expected.

There is no indication in the methods that the analysis aimed to identify gender differences, yet all results are presented separately for men and women.

Is there any discussion on the impact of HR or EtCO₂ during the 0.05 Hz sit-stand maneuvers?

Figure 1B - What is the rationale for showing the heart, Circle of Willis, and both arteries and veins? What is represented in the box labeled "microvascular function"?

Minor point - the most recent literature indicates that the official acronym for CBFV is CBv:
<https://pubmed.ncbi.nlm.nih.gov/35839156/>

Referee #2:

The study proposes a modeling approach to explain the link between the mean arterial pressure (MAP) oscillations and cortical oxygenation by measuring MAP, mean cerebral blood velocity (MCBv) via transcranial Doppler ultrasound (TCD) and oxygenated hemoglobin (O₂Hb) via near-infrared spectroscopy (NIRS) at supine resting and during sit-to-stand maneuver forcing MAP oscillations at 0.05 Hz in healthy individuals. The approach exploits a transfer function (TF) approach estimating the influence of MAP on MCBv, that of MCBv on O₂Hb and a serial model conceptualizing the link from MAP to O₂Hb. Squared coherence was assessed to test the validity of the TF estimates.

The study is very interesting and insightful. The manuscript is well-written and methodologically robust. Some modeling aspects require more careful discussion and analysis.

1) TF approach requires linearities of the input and output and their relationship. This hypothesis was not tested, and this is problematic because a reader might expect that cerebral autoregulation is a nonlinear mechanism and as such at least MCBv might exhibit nonlinear components. When explicitly tested the presence of nonlinear dynamics in MAP and MCBv was not detected (see A. Porta et al, PLoS ONE 15, e0243869, 2020). Some comments are needed in discussion to make the reader aware of the necessity of testing this hypothesis.

2) Analogous approach was applied to decompose the baroreflex into a fast neural arc and a slow mechanical arc (see Y. Ikeda et al, Am J Physiol, 271, H882-890, 1996). This approach could be mentioned to make more complete the Section "Introduction".

3) During orthostatic challenge an increase of the MCBv-MAP coupling strength is usually detected (see A. Porta et al, Eur Phys J Spec Top, 2025, doi: 10.1140/epjs/s11734-025-02074-0 and references therein). Discussion should be enlarged to more deeply support that results of the proposed modelling decomposition are in line with literature.

4) The approach is fully open loop. However, some studies suggested that the MCBv-MAP relationship should be modelled

via closed-loop approaches (see e.g., V. Bari et al, *Physiol Meas* 38: 976-991, 2017; S Saleem et al, *Am J Physiol Regul Integr Comp Physiol* 315: R484-R495, 2018). This holds in applications suggested as possible future targets of the present approach in the Section "Key points" (see F. Gelpi et al, *J Appl Physiol* 139(2), 341-354, 2025). In this case of the closed loop relationship results might have a more complex and less straight interpretation given that both feedforward and feedback pathways should be considered. Discussion should be enlarged to cover this issue.

5) TF analysis requires testing the significance of the link between the input and the output and this might be particularly problematic because of the structure of the cascade model. This aspect is evident in Figs.2. The percentage of subjects rejecting the null hypothesis of coupling in any experimental condition should be reported.

6) Several methods are present literature that do not assume the constancy of the threshold for the significance of the squared coherence (see R.B. Panerai et al. *J Physiol Meas* 39(12):125006, 2018; A Porta et al, *Med Biol Eng Comput* 61(12):3141-3157, 2023). The authors should discuss whether the use of a constant value for the threshold of significance (i.e., 0.34) regardless of the experimental condition might have biased conclusions.

7) Recordings seem to be longer than the analyzed period, especially at supine resting. The authors should indicate the logic behind the selection of the frame and whether conclusions might be influenced by the arbitrary selection of the period of analysis.

8) Usually, a much higher resampling rate was chosen (i.e., 4 Hz). The authors should discuss their choice.

9) The quality of figures 2 and 3 should be improved by reporting dispersion about the group average. In addition, standard deviation should be utilized instead of standard error. The threshold of significance of squared coherence should be reported as well. The method for the computation of the threshold of significance should be mentioned explicitly in the figure caption.

10) Conclusions held on healthy subjects but the possibility to exploit the approach over pathological individuals is not proven. This issue should be better emphasized, and future tests should be highlighted in an appropriate section.

11) Please check line 577

END OF COMMENTS

Dear Prof. Eleonora Grandi,
Senior Editor,
The Journal of Physiology

Thank you very much for considering our manuscript entitled: "**A cascade model of dynamic cerebral autoregulation**". We greatly appreciate the constructive comments from the referees and the editor, and are pleased to have the opportunity to revise our manuscript with those concerns. We have responded to them in a point-by-point fashion, with changes underlined in the revised manuscript. A clean version with no tracked changes is also attached as requested.

Sincerely,

Rong Zhang, Ph.D.

Institute for Exercise and Environmental Medicine,
Texas Health Presbyterian Hospital Dallas
7232 Greenville Ave, Dallas, TX 75231
Telephone: +1 (214) 345-4670
Fax: +1 (214) 345-4618
E-mail: RongZhang@TexasHealth.org

Responses to comments from the editor and referees.

NOTE: Line numbers refer to the submitted manuscript file; minor differences may occur in the system-generated merged PDF.

Reviewing Editor:

Ethics Concerns:

Ethics approved, number missing

RESPONSE: We have added the registered number of IRB (STU 102010-069) for the present study (Line 138).

Comments to the Author:

The manuscript has now been reviewed by 2 expert referees. In view of referees' comments, your manuscript would not be sufficient for publication in its current form. Please consider submitting a revised version for further consideration.

RESPONSE: We appreciate the opportunity to revise our manuscript. We have carefully addressed all comments in the revised version for further consideration.

Senior Editor:

Comments for Authors to ensure the paper complies with the Statistics Policy:

Please ensure the paper complies with the Journal of Physiology's Statistics Policy.

RESPONSE:

We have ensured that the revised manuscript compiles with the statistics policy for Journal of Physiology. In addition, we have added a data availability statement to the manuscript (Lines 548-550).

Comments to the Author:

I concur with the Reviewing Editor's recommendation. Please carefully revise the manuscript to address all the reviewer comments to clarify and discuss several of the modeling aspects and deepen the analysis.

RESPONSE: We have carefully revised our manuscript to address all referee comments, with particular attention to clarifying and discussing the modeling aspects. We submit the revised version for further consideration.

Referee #1:

Comment #1: Please add a rationale for using retrospective data instead of prospective data. There is no information on patient recruitment, and the main text does not provide the number of the Bioethical Committee approval.

RESPONSE: We thank the referee #1 for raising this valid point. The present study is a retrospective analysis of data obtained from a previously conducted investigation of normal aging and brain-vascular function, in which healthy volunteers were recruited using community advertisements. Because the primary objective of the work was theoretical modeling, we used this previously collected, well-characterized dataset, as now explicitly stated in the Methods. We have clarified the retrospective nature of the analysis, described participant recruitment, added two relevant references from the same study setting (Lines 130-133), and included the Institutional Review Board approval number (STU 102010-069) in the revised manuscript (Line 138).

Comment #2: "High temporal resolution" - could you provide the exact value?

RESPONSE: We have clarified the temporal resolution of the NIRO system was configured to output data at 5 Hz, and the term "high temporal resolution" has been removed to avoid ambiguity (Lines 144-146).

Comment #3: Why was only unilateral TCD used? Shouldn't the choice depend on the dominant brain hemisphere? What was the sampling frequency?

RESPONSE: We agree that bilateral TCD measurement would be preferable for assessing hemispheric asymmetries. In the present study, CBFV was measured unilaterally in the right MCA in healthy participants as part of the original data collection protocol in healthy participants. The primary objective of the current analysis was to model global pressure-flow-oxygenation dynamics rather than hemisphere-specific regulation. We acknowledge that unilateral measurements preclude direct assessment of left-right asymmetries or differential neurovascular coupling, and this limitation is now explicitly stated in the revised manuscript (Lines 490-493). There, we also note that bilateral concordance of MCA flow velocity has been demonstrated in healthy adults both at rest and during non-lateralizing physiological stimuli such as exercise, supporting the use of unilateral measurements in this context. Regarding temporal resolution, the Doppler signal was acquired at a sampling rate of 100 Hz. This information has now been clarified in the Methods section. (Lines 165-166)

Comment #4: Which type of Finapres did you use-Finapres NOVA or Finapres MIDI?

RESPONSE: We used the Finapres 2300 (Ohmeda Monitoring Systems, Englewood, CO, USA). We have added the model details in the revised manuscript (Line 168).

Comment #5: What does this mean: "with the NIRO device providing external output at 5 Hz"?

RESPONSE: We have revised the Methods to clarify the meaning of the NIRO output rate. In the present study, the NIRO system was configured to output NIRS-derived variables (e.g., TOI, nTHI) at 5 Hz. These analog outputs were digitized by the BIOPAC system at 250 Hz for synchronization with other physiological signals for offline analysis. (Lines 144-146, and 174–176).

Comment #6: Please provide a reference here: "Participants were instructed to refrain from high-intensity exercise, caffeine, and alcohol for at least 24 hours prior to testing."

RESPONSE: We have added a reference (Panerai et al., 2023) to the CARNet white paper emphasizing pre-test standardization (including abstinence from caffeine, alcohol, and strenuous exercise) to minimize confounding influences on cerebral and systemic hemodynamics (Lines 181-182).

Comment #7: Did you control only EtCO₂, or tidal volume as well?

RESPONSE: Neither EtCO₂ nor tidal volume was experimentally controlled during rest or sit–stand maneuvers; participants breathed spontaneously under both conditions. The power spectra of beat-to-beat changes in HR and breath-by-breath EtCO₂ during rest and sit–stand maneuvers are now presented in the Appendix, Figure A1. We have added a corresponding limitation in the Discussion noting that potential influences of HR and EtCO₂ on the cascade modeling cannot be excluded (Lines 481-85). Please also see the response to Comment #19.

Comment #8: How were data averaged across the cardiac cycle? Please describe this sentence in more detail: "Beat-to-beat values of MAP, CBFV, TOI, and nTHI were analyzed using AcqKnowledge software (BIOPAC Systems, Goleta, CA, USA) and averaged across cardiac cycles (Zhang et al., 1998)."

RESPONSE: We have revised the manuscript to clarify the cardiac cycle averaging procedure. Specifically, R-wave peaks were detected from the ECG, each cardiac cycle was defined by consecutive R–R intervals, and beat-to-beat values of MAP, CBFV, TOI, and nTHI were obtained by integrating each signal over the corresponding R–R interval to yield a single value per cycle. This procedure is now described in the Methods section (Lines 194-199)

Comment #9: Why is there a '#' symbol before (1) in the formula?

RESPONSE: The symbols noted by the reviewer were hidden field codes associated with equation numbering and alignment that became visible during PDF conversion. These field codes have now been removed and the equations reformatted accordingly.

Comment #10: Please add a reference to the latest CARNET paper in the "Transfer function analysis" section.

RESPONSE: Thank you for this suggestion. The CARNet consensus white paper (Panerai et al., 2023) is cited in the Transfer Function Analysis section (Lines 222, 242, and 251).

Comment #11: It should probably be "cross-spectrum" instead of "crossspectrum."

RESPONSE: We have corrected “crossspectrum” to “cross-spectrum.” Similarly, “auto-spectrum” has been corrected where applicable throughout the manuscript.

Comment #12: "The phase describes the temporal offset between the input and output (reflecting the time constant and/or delay of the vascular adjustment in response to changes in blood pressure or blood flow). The coherence represents the consistency of a linear relationship between the input and output across frequencies." - This is correct, but a general interpretation of phase shift and gain could also be helpful here.

RESPONSE: Thank you for this suggestion. We have revised and expanded the Transfer Function Analysis section to interpret the gain, phase, and coherence, that is, gain reflects attenuation, phase captures temporal lead–lag relationships, and coherence assess the reliability of gain and phase estimation (Line 224-232).

Comment #13: In all formulas (3a, 3b, 3c), the names of Spv, Svc, etc., should be provided.

RESPONSE: We have now clarified all spectral terms used in Equations (3a–c) in the revised manuscript (Lines 237-239)

Comment #14: "To avoid 'phase wrapping', negative phase values below 0.10 Hz during rest were excluded from the phase calculation." - Please comment on how often such values were observed.

RESPONSE: We thank the reviewer for highlighting this point. We have now clarified the frequency-dependent occurrence of negative phase values in the Results and now report the proportion of subjects exhibiting negative MAP–CBFV phase across frequencies. Specifically, negative phase values were most prevalent at the lowest frequencies (~40% of subjects at 0.0078 Hz) and became rare near 0.10 Hz (<5%), with the full distribution provided in the Appendix, Table A1 (Lines 302-304).

Comment #15: "0.05 Hz during repeated sit-stand maneuvers." - Is there any range around this frequency? Did you check whether there were fluctuations in the spectrum?

RESPONSE: During repeated sit-stand maneuvers, transfer function analyses was performed over the 0-0.5Hz frequency range. To represent the periodic response at 0.05 Hz, the power spectral density and transfer function values at the two adjacent frequency bins (0.046875 and 0.0546875 Hz) were averaged, consistent with prior studies (Tomoto et al., 2021; Tarumi et al., 2022). We have clarified this procedure in the revised Methods section (Lines 261-264). In addition, the full TFA spectra across 0-0.5 Hz are now shown in the Appendix Figure A2 to illustrate spectral variability around the target frequency.

Comment #16: I suggest using either mean (plus minus) SD with parametric tests or median (plus minus) IQR with non-parametric tests. Mixing the two approaches results in different statistical power.

RESPONSE: In accordance with the referee #1's suggestion, we have revised the tables to report data as mean (\pm SD) and applied parametric statistical testing with unpaired Student's t-tests. This change is reflected in the Methods (Line 275) and Tables I–III.

Comment #17: "Men had significantly greater height, weight, and BMI than women." - This finding is rather expected.

RESPONSE: We agree with the referee #1 and have removed this sentence from the revised manuscript.

Comment #18: There is no indication in the methods that the analysis aimed to identify gender differences, yet all results are presented separately for men and women.

RESPONSE: Indeed the primary analyses were not designed to test sex differences, therefore, we have revised the Tables I-III to also present the results for all subjects combined (ALL), reflecting the main focus of the study. Sex-stratified values are retained to facilitate comparison with prior literature which showed that several cerebrovascular indices (e.g., CBFV and TOI) differ by sex.

Comment #19: Is there any discussion on the impact of HR or EtCO₂ during the 0.05 Hz sit-stand maneuvers?

RESPONSE: The average HR under rest and sit-stand conditions has been added to Table I. As expected, HR increased during sit-stand maneuvers ($p < 0.001$) while EtCO₂ did not differ between rest and sit-stand conditions ($p = 0.074$). The PSDs of HR and EtCO₂ (Appendix Figure A1 in this revision) showed spectral patterns similar to those of MAP, CBFV, and O₂Hb_{SRS} under both rest and sit-stand conditions. Since beat-to-beat changes in HR and breath-by-breath changes in EtCO₂ likely correlate with those of MAP, CBFV, and O₂Hb_{SRS}, their potential confounding effects on the estimation of dCA and MF cannot be excluded. This limitation has now been noted in the Discussion section (Lines 481-485).

Comment #20: Figure 1B - What is the rationale for showing the heart, Circle of Willis, and both arteries and veins? What is represented in the box labeled "microvascular function"?

RESPONSE: This panel is intended as a conceptual schematic illustrating the serial organization of the cerebrovascular cascade modeled. As clarified in the revised figure legend, the heart, middle cerebral artery, and capillary bed schematically represent the source of systemic arterial pressure, the macrovascular pressure-flow compartment $H_1(f)$, and microvascular oxygenation compartment $H_2(f)$, respectively. We also clarified that the "microvascular function" box is intended as a conceptual illustration of the neurovascular unit rather than a depiction of specific cellular mechanisms included in the model (Lines 807-810). No changes were made to the model itself.

Comment #21: Minor point - the most recent literature indicates that the official acronym for CBFV is CBv: <https://pubmed.ncbi.nlm.nih.gov/35839156/>

RESPONSE: We appreciate the reviewer's reference to the recent discussion regarding TCD terminology. We carefully considered adoption of cerebral blood velocity (CBv); however, we elected to retain cerebral blood flow velocity (CBFV) for consistency with its longstanding usage in cerebral hemodynamics (*Nichols et al., McDonald's Blood Flow in Arteries, 7th ed., 2022*), Doppler ultrasound standards (*Mitchell et al., J Am Soc Echocardiogr, 2019*), and the physical sciences, where flow velocity is a well-established term in engineering and fluid mechanics (e.g., *White, Fluid Mechanics, 7th ed., 2011*). Although Doppler measures red blood cell velocity, these cells act as indicators of bulk blood flow, and the signal is conventionally interpreted as flow velocity in both physiological and clinical contexts. In addition, because CBV is widely used to denote cerebral blood volume, adoption of CBv could introduce ambiguity. We have added a clarifying statement in the Methods specifying that CBFV refers to Doppler-derived velocity in the middle cerebral artery and does not imply direct measurement of volumetric cerebral blood flow (Lines 160-161).

Referee #2:

General Comment: The study proposes a modeling approach to explain the link between the mean arterial pressure (MAP) oscillations and cortical oxygenation by measuring MAP, mean cerebral blood velocity (MCBv) via transcranial Doppler ultrasound (TCD) and oxygenated hemoglobin (O₂Hb) via near-infrared spectroscopy (NIRS) at supine resting and during sit-to-stand maneuver forcing MAP oscillations at 0.05 Hz in healthy individuals. The approach exploits a transfer function (TF) approach estimating the influence of MAP on MCBv, that of MCBv on O₂Hb and a serial model conceptualizing the link from MAP to O₂Hb. Squared coherence was assessed to test the validity of the TF estimates.

The study is very interesting and insightful. The manuscript is well-written and methodologically robust. Some modeling aspects require more careful discussion and analysis.

RESPONSE: We thank Referee #2 for the positive evaluation of our study and for the constructive comments on the modeling framework. We have revised the manuscript to clarify and strengthen the presentation and interpretation of the cascade model.

Comment #1: TF approach requires linearities of the input and output and their relationship. This hypothesis was not tested, and this is problematic because a reader might expect that cerebral autoregulation is a nonlinear mechanism and as such at least MCBv might exhibit nonlinear components. When explicitly tested the presence of nonlinear dynamics in MAP and MCBv was not detected (see A. Porta et al, PLoS ONE 15, e0243869, 2020). Some comments are needed in discussion to make the reader aware of the necessity of testing this hypothesis.

RESPONSE: The reviewer raises an important point regarding the linearity assumption underlying transfer-function analysis. We agree that a discussion of potential non-linear properties is important for the rigor of the manuscript and the validity of the results.

In response to this suggestion, we have added a section to the Discussion (under "Limitations") to address this issue. Specifically, we acknowledge that while formal non-linear analyses were not performed in this study, recent work (Porta et al, 2020) suggested that non-linearity presented in

the MAP-CBFV relationship are likely to be minimal even under clinical conditions. We also clarified that low coherence observed in the VLF band may be attributed to unaccounted physiological variables (like CO₂) rather than a breakdown of linearity itself (Panerai et al., 2006) (Peng et al., 2008, 2010). Finally, we highlight that the convergence of our cascade model estimates with our empirical findings during the large forced oscillations of the sit-stand maneuver suggests that linear approximation effectively captures the dominant dynamics (Lines 457-471).

Comment #2: Analogous approach was applied to decompose the baroreflex into a fast neural arc and a slow mechanical arc (see Y. Ikeda et al, Am J Physiol, 271, H882-890, 1996). This approach could be mentioned to make more complete the Section "Introduction".

RESPONSE: The cited study is highly relevant to the conceptual basis of the proposed cascade model. We have now added the study by Ikeda et al. (Am J Physiol, 1996) to the Introduction. This work demonstrated that the arterial baroreflex can be decomposed into neural and peripheral arcs and modeled as a series system, enabling physiologically meaningful interpretation using transfer function analysis. This addition complements the previously cited studies by Shibata et al. and Hieda et al. and highlights established precedent for the cascade modeling of cardiovascular regulation (Lines 117-119).

Comment #3: During orthostatic challenge an increase of the MCBv-MAP coupling strength is usually detected (see A. Porta et al, Eur Phys J Spec Top, 2025, doi: 10.1140/epjs/s11734-025-02074-0 and references therein). Discussion should be enlarged to more deeply support that results of the proposed modelling decomposition are in line with literature.

RESPONSE: We thank the referee #2 for highlighting the orthostatic challenge literature and directing us to the recent work by Porta et al. (2025). In response, we have expanded the Discussion to more explicitly situate our sit-stand findings within the broader literature demonstrating increased MAP-CBFV coupling during orthostatic stress. Specifically, we now note that enhanced pressure-flow coupling during standing or repeated sit-stand maneuvers has been consistently reported across linear transfer-function analyses, wavelet-based methods, and nonlinear directionality frameworks (Lines 464-471). We further discuss how these findings align with the sequential, feedforward structure assumed in the present cascade model (Lines 476-480).

Comment #4: The approach is fully open loop. However, some studies suggested that the MCBv-MAP relationship should be modelled via closed-loop approaches (see e.g., V. Bari et al, Physiol Meas 38: 976-991, 2017; S Saleem et al, Am J Physiol Regul Integr Comp Physiol 315: R484-R495, 2018). This holds in applications suggested as possible future targets of the present approach in the Section "Key points" (see). In this case of the closed loop relationship results might have a more complex and less straight interpretation given that both feedforward and feedback pathways should be considered. Discussion should be enlarged to cover this issue.

RESPONSE: Referee #2 raises an important point regarding the potential closed-loop nature of the MAP-CBFV relationship. In response, we have expanded the Discussion to acknowledge studies suggesting bidirectional MAP-CBFV interactions (Bari et al., 2017; Saleem et al., 2018), while noting that it remains unclear whether these observations reflect true physiological feedback

mechanisms or statistical interdependencies arising from shared cardiovascular–cerebrovascular regulation. We further clarify that the present cascade framework models $BP \rightarrow CBFV \rightarrow O_2Hb$ dynamics as an open-loop series system focusing on feed-forward transmission of pressure fluctuations, which is particularly appropriate under the forced oscillatory conditions used here (Porta et al., 2025). We also note that extending the framework to closed-loop formulations represents an important direction for future work (Lines 472–480).

Comment #5: TF analysis requires testing the significance of the link between the input and the output and this might be particularly problematic because of the structure of the cascade model. This aspect is evident in Figs.2. The percentage of subjects rejecting the null hypothesis of coupling in any experimental condition should be reported.

RESPONSE: We have revised the manuscript to explicitly report the percentage of subjects rejecting the null hypothesis of no coupling for each cascade pathway and experimental condition. Coupling significance was assessed using the critical coherence thresholds derived according to the updated CARNet recommendations (Panerai et al., 2023). Significant coupling was defined when coherence exceeded 0.34, corresponding to the 95% confidence limit for five overlapping spectral windows employed in the presented study. This methodology has been added to the Methods section (Lines 265-271).

Subject-level detection rates are summarized in Table V. Briefly, during rest, significant linear coupling was observed in 90.2% of subjects for H_1 , 61.0% for H_2 , and 41.6% for H_0 , whereas during sit–stand maneuvers these proportions increased to 100% for H_1 , H_2 and H_0 at 0.05 Hz. These results and their interpretation have been added to the Results and Discussion sections. (Lines 338-342, and Lines 399-407, respectively).

Comment #6: Several methods are present literature that do not assume the constancy of the threshold for the significance of the squared coherence (see R.B. Panerai et al. J Physiol Meas 39(12):125006, 2018; A Porta et al, Med Biol Eng Comput 61(12):3141-3157, 2023). The authors should discuss whether the use of a constant value for the threshold of significance (i.e., 0.34) regardless of the experimental condition might have biased conclusions.

RESPONSE: In this study, the significance threshold for coherence was determined according to the methodological recommendations summarized in the CARNet white paper (Panerai et al., 2023), which derived critical coherence limits from Monte Carlo simulations of independent white Gaussian noise under defined spectral estimation conditions. For the spectral parameters used in our analysis (256-point Hanning windows with 50% overlap applied to ~5-minute recordings), the resulting spectral estimation produced five overlapping Welch segments, corresponding to a 95% confidence limit for a magnitude-squared coherence of 0.34.

Because the same data length and spectral estimation procedure were applied to all subjects and experimental conditions, the corresponding coherence threshold is determined by the analysis parameters rather than arbitrarily fixed, and therefore does not introduce differential bias between rest and sit–stand analyses.

We have clarified this methodology and its rationale in the revised Methods (Lines 265-271), summarized relevant results in the Results (Lines 338-342), and discussed its limitations in the Discussion (Lines 442-456) sections. Additionally, we have added a brief discussion of complementary directionality-aware approaches for considering closed-loop formulations, following Porta et al. (2023).

Comment #7: Recordings seem to be longer than the analyzed period, especially at supine resting. The authors should indicate the logic behind the selection of the frame and whether conclusions might be influenced by the arbitrary selection of the period of analysis.

RESPONSE: The present study is based on previously acquired datasets in which experimental protocols were designed to ensure physiological stabilization. For the supine resting condition, analyses were performed on the last 5-minutes of a physiologically stable segment selected from the 10-minute rest period, rather than on the full recording. The analyzed segment was selected after completion of the stabilization phase based on signal stability and absence of artifacts, in order to minimize transient effects and ensure steady-state conditions. This selection was therefore protocol-driven rather than arbitrary and has now been clarified in the Methods section (Lines 183-185).

The analyzed data lengths were 309 ± 40 s for the resting condition and 295 ± 10 s for sit-stand maneuvers. Previous studies indicate that transfer function estimates of dCA are relatively robust to differences in recording length when ≥ 3 minutes of data are available (*Physiol Meas* 2019;40:085002; *Biomed Res Int.* 2018; 2018:7803426). Therefore, the small variability in analyzed data length in the present study is unlikely to have significant influences on the TFA results.

Comment #8: Usually, a much higher resampling rate was chosen (i.e., 4 Hz). The authors should discuss their choice.

RESPONSE: In the present study, all signals were resampled at 2 Hz. This rate provides a Nyquist frequency of 1 Hz, which is sufficient to capture cerebrovascular dynamics within the analyzed range (≤ 0.5 Hz) without aliasing. The choice of 2 Hz is consistent with our prior studies using similar experimental paradigms and datasets (*Tarumi et al., 2014; Xing et al., 2017; Tarumi et al., 2022*), thereby ensuring methodological continuity and comparability. Although higher resampling rates (e.g., 4 Hz) are recommended, particularly in the settings with elevated heart rates ($> 120/\text{min}$), such conditions were not present in our healthy cohort. We have added a clarifying statement in the Methods to explain this rationale (Lines 210–212).

Comment #9: The quality of figures 2 and 3 should be improved by reporting dispersion about the group average. In addition, standard deviation should be utilized instead of standard error. The threshold of significance of squared coherence should be reported as well. The method for the computation of the threshold of significance should be mentioned explicitly in the figure caption.

RESPONSE: We thank referee #2 for this suggestion to improve the figures. To better represent dispersion around the group average, Figures 2 and 3 have been revised to display the group-averaged frequency spectra as solid lines, with the variability across subjects illustrated by shaded gray bands corresponding to the 95% confidence interval.

Although standard deviation (SD) is reported for summary statistics, SD-based envelopes were not used for graphical display because they can exceed the theoretical upper bound of coherence (>1), resulting in misleading visual representations. Accordingly, SD values are reported in Tables II and III, while confidence intervals are used for figures which provide the precision of spectral estimation.

In addition, the significance thresholds for squared coherence are now shown separately in the Appendix Figure A3 to preserve clarity of the main spectral figures. The number and percentage of subjects exceeding the critical coherence threshold are now reported in Table V.

Comment #10: Conclusions held on healthy subjects but the possibility to exploit the approach over pathological individuals is not proven. This issue should be better emphasized, and future tests should be highlighted in an appropriate section.

RESPONSE: We agree that the present conclusions are derived from healthy participants and that applicability to pathological populations has not yet been established. In response, we have revised the Discussion to more clearly emphasize this limitation and to state that the findings should be interpreted within a healthy physiological context. We now explicitly note that the generalizability of the cascade model to aging or disease states remains to be determined and requires future studies (Lines 512-515).

We have also modified the “Key Points” section to clarify that application of the framework to aging and cerebrovascular disease represents a potential direction for future studies (Lines 44–45).

Comment #11: Please check line 577

RESPONSE: We have removed this citation formatting error in the revised manuscript.

Dear Professor Zhang,

Re: JP-RP-2026-290519R1 "A Cascade Model of Dynamic Cerebral Autoregulation" by Takuya Kurazumi, Kartavya Sharma, Ricardo Wennekers, Tsubasa Tomoto, Danilo Cardim, Junyeon Won, John Ashley, Jurgen A.H.R. Claassen, and Rong Zhang

We are pleased to tell you that your paper has been accepted for publication in The Journal of Physiology.

Yours sincerely,

Eleonora Grandi
Senior Editor
The Journal of Physiology

IMPORTANT POINTS TO NOTE FOLLOWING ACCEPTANCE OF YOUR PAPER:

- **IMPORTANT NOTICE ABOUT OPEN ACCESS:** To assist authors whose funding agencies mandate immediate public access to published research findings, The Journal of Physiology allows authors to pay an Open Access (OA) fee to have their papers made freely available immediately on publication.

The Corresponding Author will receive an email from Wiley with details on how to register or log in to Wiley Authors where you will be able to place an order.

- You can check if your funder or institution has a Wiley Open Access Account here:
<https://authors.wiley.com/author-resources/Journal-Authors/open-access/author-compliance-tool.html>

- You can help your research get the attention it deserves! Check out Wiley's free Promotion Guide for best-practice recommendations for promoting your work at: www.wileyauthors.com/eoo/guide. You can learn more about Wiley Editing Services which offers professional video, design, and writing services to create shareable video abstracts, infographics, conference posters, lay summaries, and research news stories for your research at: www.wileyauthors.com/eoo/promotion.

- If you would like to receive our 'Research Roundup', a monthly newsletter highlighting the cutting-edge research published in The Physiological Society's family of journals (The Journal of Physiology, Experimental Physiology, Physiological Reports, The Journal of Nutritional Physiology and The Journal of Precision Medicine: Health and Disease), please click this link, fill in your name and email address and select 'Research Roundup':
<https://www.physoc.org/journals-and-media/membernews>

EDITOR COMMENTS

Reviewing Editor:

This study introduces a modeling approach to explain how mean arterial pressure oscillations relate to cortical oxygenation,

through multimodal physiological measurements. Using a transfer function analysis, this modeling presents an estimation of the "cascade" effects from mean arterial pressure up to oxyhemoglobin.

The authors have addressed all the concerns presented by the reviewers. The manuscript is suitable for publication.

Senior Editor:

Thank you for thoroughly addressing all comments. Congratulations!

REFeree COMMENTS

Referee #1:

I have carefully reviewed the responses and found them satisfactory. I would like to thank the Editor again for the opportunity to review this manuscript.

Referee #2:

The manuscript has been improved. The authors replied satisfactorily to all my issues and took into account the suggestions given. I have no additional comments.